# Clipping Low-Probability Tokens in SFT Yields a Generalizable Initialization for RL

Tian-Shuo Liu [1 2 *]   Chengxing Jia [1 *]   Haoyu Liu [1 *]   Pengyuan Wang [1 2]   Shiyuan Zhang [1]   Jie Fu [3]   Yang Yu [1 2 †]

## Abstract

Supervised Fine-Tuning (SFT) is a critical step for adapting Large Language Models (LLMs) to specialized domains, often serving as an initialization for subsequent reinforcement learning (RL). However, SFT can overfit a small set of expert data, harming generalization and eroding prior knowledge. This can limit downstream RL, which benefits from a strong, generalizable initialization for exploration. Here, we demonstrate that prior knowledge degradation primarily results from tokens in the expert data to which the base model assigns low probability. Specifically, these low-probability tokens represent a significant deviation from the model's current prior knowledge. Due to the nature of the log-likelihood objective, they produce larger gradient magnitudes, which speed up adaptation to the new data but degrade generalization. In this paper, we study the token-wise clipping strategy, a commonly used trust-region method for bounding per-token updates. We find that it reshapes token-level learning priorities, promoting more progressive adaptation that fits the new data while preserving general abilities. Compared with standard SFT, clipping low-probability tokens reduces out-of-distribution forgetting by 11.54% and improves final RL performance by 7.09% across the agentic benchmarks. Moreover, latent-space analysis shows smaller representational drift under clipping, indicating that it provides a generalizable initialization.

## 1. Introduction

Large Language Models (LLMs) have emerged as a powerful foundation for tackling complex reasoning and decision-making tasks (OpenAI et al., 2024; Team et al., 2025; Wang et al., 2026). Although LLMs are routinely pre-trained on billions of tokens, it is insufficient to produce models that are adept at specialized downstream tasks or capable of robust, multi-step reasoning (DeepSeek-AI et al., 2025; Liu et al., 2021). Therefore, post-training, which includes Supervised Fine-Tuning (SFT) and Reinforcement Learning (RL), is the critical stage for continually improving LLMs and enabling the acquisition of new abilities (Kumar et al., 2025; Zhang et al., 2024a). Within this paradigm, SFT adapts a model to a new domain by training it on a curated set of expert data. This process instills task-specific behaviors and serves as an essential cold-start, namely the initialization for the subsequent RL phase (Ouyang et al., 2022; Qwen et al., 2025; Yu et al., 2025a; Wang et al., 2025).

Although this process is adept at cloning a specific behavioral policy (DeepSeek-AI et al., 2025; Wei et al., 2025), the model's tendency to memorize these static traces leads to impaired generalization and the catastrophic forgetting of pre-existing knowledge (Chu et al., 2025; Wu et al., 2025b; Shenfeld et al., 2025). This occurs as SFT inadvertently alters the model's internal representations, causing an erosion of the foundational knowledge acquired during pre-training. This degradation is particularly detrimental for the subsequent RL phase. A flawed cold-start means initializing the RL agent in a less generalizable and knowledgeable state, which limits the generation of useful exploratory experiences and imposes a ceiling on its performance (Huan et al., 2025; Zhao et al., 2025). Consequently, a fundamental question arises:

*What mechanism in SFT induces forgetting of prior knowledge, and can we mitigate it to obtain an initialization that preserves general abilities?*

In response to this question, we analyze the training dynamics of SFT on agentic benchmarks (Luo et al., 2025a; Shridhar et al., 2021), where a strong cold-start is essential (Shang et al., 2025). We observe that degradation is fastest at the very beginning of fine-tuning. Out-of-distribution perfor-

---

[*]Equal contribution [†]Corresponding author [1]National Key Laboratory for Novel Software Technology, Nanjing University, Nanjing, China & School of Artificial Intelligence, Nanjing University, Nanjing, China [2]Polixir Technologies [3]IQuest Research. Correspondence to: Yang Yu <yuy@nju.edu.cn>.

*Proceedings of the 43rd International Conference on Machine Learning*, Seoul, South Korea. PMLR 306, 2026. Copyright 2026 by the author(s).

mance drops sharply within the first updates, while the overall gradient norm is already high and token probabilities change rapidly, indicating aggressive early adaptation to the expert data. By investigating the relationship between target-token probability and gradient norm, we find that this early training signal is dominated by a small subset of targets that the base model assigns very low probability. We refer to such targets as off-policy tokens, meaning that they are unlikely under the model's current predictive distribution. Under the log-likelihood objective, these off-policy targets receive disproportionately large updates. Although they accelerate fitting to the expert data, they also exacerbate forgetting and degrade generalization. These findings motivate further study of whether token-level constraints can make SFT more progressive, allowing the model to absorb new targets while minimizing forgetting of prior knowledge.

Motivated by this diagnosis, we further examine token-wise clipping, which is widely used in trust-region methods to bound per-token updates (Schulman et al., 2015; 2017). We adopt a simple instantiation of this clipping strategy and refer to it as Off-Policy token Clipped SFT (OPC-SFT). We next conduct experiments on the agentic benchmarks ALFWorld and ScienceWorld. Compared with standard SFT, we find that OPC-SFT reduces out-of-distribution forgetting by 11.54% and improves final downstream RL performance by 7.09%. We further analyze the training dynamics and find that OPC-SFT induces substantially smaller representational drift in latent space. We also observe smoother probability trajectories under OPC-SFT, suggesting that clipping suppresses the most drastic early updates. We isolate the importance-weighting component implicit in our clipping rule and perform sensitivity analyses over the main hyperparameters. These studies show that clipping low-probability tokens during SFT provides a practical and robust way to mitigate forgetting and improve generalization. Finally, experiments on math reasoning tasks indicate that the benefits of OPC-SFT are more pronounced when the SFT data is substantially off-policy. The code for our experiments is available at https://github.com/LTSure/on-policy-cold-start.

## 2. Preliminaries

### 2.1. A Two-Stage Training Paradigm for LLMs

A sequential 'SFT-then-RL' pipeline has emerged as the standard, adopted by numerous state-of-the-art open-source models (Yoshihara et al., 2025). In this two-stage approach, SFT serves as a cold-start stage before RL. It initializes the model by imitating expert demonstrations, providing a starting point for subsequent RL exploration. Let $\pi_\theta(y \mid x)$ denote the conditional generative distribution modeled by an LLM with parameters $\theta$. Given an input prompt $x$, the model generates an output sequence $y = (y_1, \ldots, y_T)$ au-

toregressively. Concretely, SFT is performed on a small set of high-quality supervised demonstrations $\mathcal{D} = \{(x, y^*)\}$ by minimizing the negative log-likelihood objective:

$$\mathcal{L}_{\text{SFT}}(\theta) = \mathbb{E}_{(x,y^*)\sim\mathcal{D}}[-\log \pi_\theta(y^* \mid x)]. \quad (1)$$

This SFT phase yields an LLM model $\pi_0$, which serves as the starting point for reinforcement learning.

SFT is an effective paradigm that can rapidly improve in-domain performance. However, when the demonstration set $\mathcal{D}$ is small and distributionally shifted, SFT can encourage memorization of the expert traces, leading to overfitting and weaker generalization on out-of-distribution inputs, which can hinder subsequent RL training.

During the RL phase, we frame sequential token-wise generation as a decision-making process. We define a state space $\mathcal{X}$ and an action space $\mathcal{Y}$. At each timestep $t$, the LLM acts as a policy $\pi : \mathcal{X} \rightarrow \Delta(\mathcal{Y})$, where $\Delta(\cdot)$ denotes the probability simplex. A state $x_t \in \mathcal{X}$ is the prompt concatenated with all previously generated tokens, and an action $y_t \in \mathcal{Y}$ corresponds to the next token to be generated. Reinforcement Learning with verifiable rewards (RLVR) optimizes this process using a reward function $R : \mathcal{X} \times \mathcal{Y} \rightarrow 0, 1$ to guide the model toward desired behaviors.

### 2.2. Token-Level Clipping Strategy

When the demonstration set $\mathcal{D}$ is small and distributionally shifted, standard SFT can induce overly aggressive updates on a subset of targets, leading to overfitting and weaker out-of-distribution generalization, which can in turn hinder subsequent RL training. There are several natural options that may mitigate this phenomenon. Decreasing the learning rate uniformly slows learning for all tokens, including those that are already well-behaved. Replay-based approaches are effective for mitigating forgetting in sequential-task continual learning (Scialom et al., 2022), but are less suitable for preserving broad pre-trained abilities, since constructing a representative replay buffer is non-trivial. Another option is to regularize distributional shift via KL-based penalties (Schulman et al., 2015), yet enforcing KL constraints can be costly and often relies on approximations that may not be robust in practice.

A simple alternative is token-level clipping, which is widely used in trust-region methods as a practical way to bound update magnitudes (Schulman et al., 2017). Token-level clipping constrains the policy update by comparing the current policy $\pi_\theta$ to the old reference policy $\pi_{\theta_{\text{old}}}$. This is typically implemented using a probability ratio $r_t(\theta)$ and an advantage estimate $\hat{A}_t$ at timestep $t$ in reinforcement learning:

$$r_t(\theta) = \frac{\pi_\theta(y_t \mid s_t)}{\pi_{\theta_{\text{old}}}(y_t \mid s_t)}. \quad (2)$$

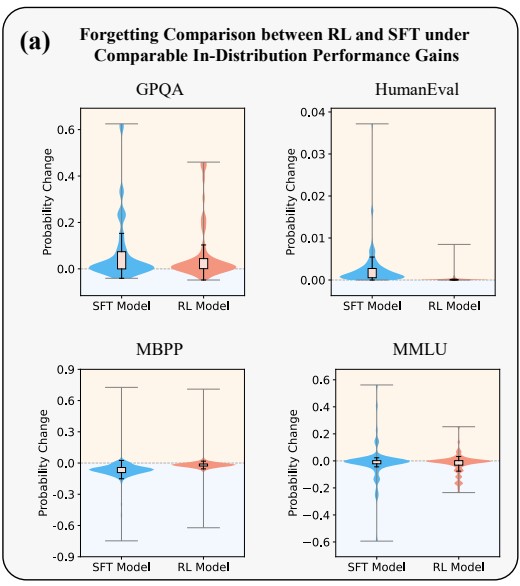
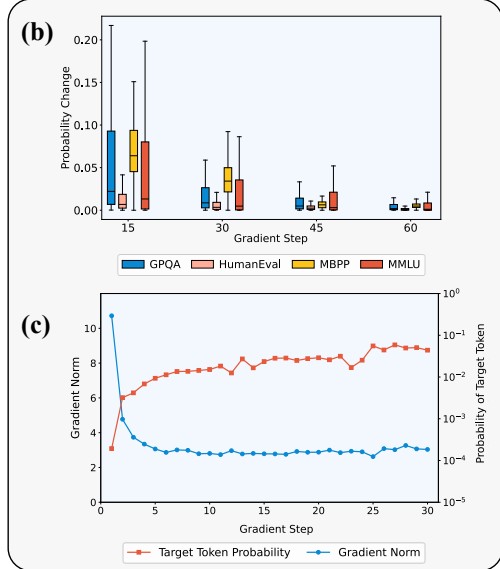

*Figure 1.* (a) Probability changes comparison between RL and SFT models after both achieving comparable in-distribution validation performance. (b) Probability changes during training. (c) Curves of gradient norm and target token probability for tokens in the bottom 1% quantile.

This ratio is then used in a clipped objective, which limits large deviations from the old policy:

$$
\begin{aligned}
\mathcal{L}_{\mathrm{CLIP}}(\theta) = \\
\mathbb{E}_t \left[ \min\left( r_t(\theta)\hat{A}_t,\ \mathrm{clip}(r_t(\theta), 1 - \epsilon, 1 + \epsilon)\,\hat{A}_t \right) \right].
\end{aligned} \tag{3}
$$

This objective provides a simple token-level constraint, which is especially effective for low-probability tokens.

# 3. Low-Probability Token Clipped Supervised Fine-Tuning

In this section, we provide an analysis of generalization degradation during the SFT process. We begin by showing that a subset of low-probability tokens induces disproportionately large gradient norms during fine-tuning. These large gradients lead to abrupt probability shifts, reflected in rapid changes in token probabilities on the target data and on out-of-distribution tasks, indicating aggressive adaptation to the new data. Such aggressive early adaptation is accompanied by forgetting of general abilities and degraded out-of-distribution generalization. Based on this insight, we adopt a clipping strategy for these low-probability tokens to test the hypothesis that selectively constraining their updates mitigates forgetting. We ground our validation in the domain of agentic tasks, specifically using the agentic benchmark (Shridhar et al., 2021). This environment is an ideal testbed because the text-based embodied task requires a cold-start for the LLM to learn the specific decision-making format, a capability usually absent during pre-training.

## 3.1. Generalization Degradation During SFT

Our investigation starts from the observation that SFT tends to reallocate probability mass toward task-specific patterns, often at the expense of general knowledge (Chu et al., 2025; Huan et al., 2025). Empirically, we post-train a warmed-up Llama3.2-3B-Instruct model on ALFWorld (Shridhar et al., 2021) with SFT and RL respectively, until the two methods achieve comparable performance on the in-distribution test set. We then evaluate knowledge retention of these models on out-of-distribution benchmarks including coding and QA tasks: GPQA (Rein et al., 2023), HumanEval (Chen et al., 2021), MBPP (Austin et al., 2021), and MMLU (Hendrycks et al., 2021a). We quantify retention by measuring probability shifts, where smaller changes indicate better knowledge preservation. Fig. 1(a) shows that SFT causes substantially larger probability shifts than RL, suggesting that SFT often learns new behaviors at the cost of disrupting prior behavior. To determine when degradation occurs, we track probability change across SFT checkpoints. Fig. 1(b) plots the incremental change between consecutive checkpoints and shows that it drops sharply from epoch 1 to epoch 2, indicating that most forgetting occurs early in SFT. It motivates us to further examine the evolution of target token probabilities and gradient norms during training. Fig. 1(c) shows that the gradient norm is highest at the start of training and drops sharply thereafter, while target tokens in the lowest 1% probability quantile simultaneously experience a rapid increase in probability. These high-magnitude gradients are, in turn, responsible for the large probability changes observed in Fig. 1(b). These results suggest that catastrophic forgetting is largely driven by a small subset of off-policy targets, to-

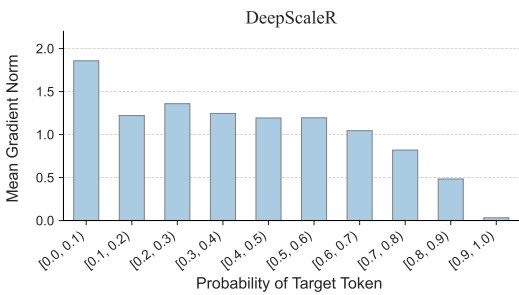 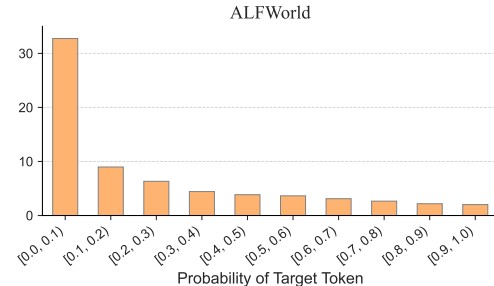

*Figure 2.* Mean gradient norm for tokens in different probability bins.

kens that are unlikely under the model's current distribution and deviate from its prior behavior. In Sec. 3.2, we test this hypothesis by linking target-token probability to gradient norms, and we examine whether selectively constraining updates from these tokens can mitigate forgetting.

### 3.2. Low-Probability Tokens and Large Gradient Norms

The standard objective for SFT is to maximize the likelihood of an expert-provided response $y^*$ given the input query $x$. This is achieved by minimizing the negative log-likelihood loss for each sample in a dataset $\mathcal{D} = \{(x, y^*)\}$:

$$\nabla_\theta \mathcal{L}_{\text{SFT}}(\theta) = -\mathbb{E}_{(x,y^*)\sim\mathcal{D}} \left[ \frac{\nabla_\theta \pi_\theta(y^* \mid x)}{\pi_\theta(y^* \mid x)} \right]. \quad (4)$$

While this objective is intuitive, it may exhibit sudden and excessively large gradient magnitudes. This phenomenon stems from the $\pi(y^*|x)$ term in the denominator. During the early cold-start phase, the model assigns very low probability to a subset of target tokens in the expert data, as shown in Fig. 1(c). When the token probability $\pi_\theta(y_t \mid x_t)$ becomes small, e.g., on the order of $10^{-4}$, the gradient magnitude can become extremely large. Our fine-grained analysis in Fig. 2 further validates that low-probability tokens are associated with much larger gradient norms. Specifically, we partition tokens into ten bins according to their model-assigned probabilities. Tokens in the lowest-probability bin $[0, 0.1)$, which we term off-policy tokens, meaning targets that are unlikely under the current model distribution, exhibit an average gradient norm of 32.75, in sharp contrast to 2.02 for high-confidence tokens in the $[0.9, 1.0]$ range. These large gradients indicate that fitting these new targets requires substantial updates, but the same updates can come at the expense of general abilities and prior knowledge retention.

### 3.3. Clipping Low-Probability Tokens

This phenomenon naturally prompts a direction: we can selectively constrain token-level updates to enforce more progressive adaptation, allowing the model to learn new behaviors while preserving prior knowledge. Straightforward

approaches are not well-suited to our setting. Decreasing the learning rate uniformly slows learning for all tokens, but it does not specifically curb the large updates induced by low-probability tokens. Over many steps, the resulting updates can still lead to the same forgetting. Global clip-by-norm also shrinks the overall step size. However, low-probability tokens can still accumulate large influence over training, albeit more slowly. Replay-based approaches are commonly used in sequential-task continual learning, yet they do not directly address preserving broad pre-trained abilities, since constructing a representative replay buffer for general capabilities is non-trivial.

Trust-region methods introduced in Sec. 2.2 provide a natural mechanism for controlling update magnitudes by introducing an old policy $\pi_{\theta_{\text{old}}}$ and limiting deviations from it. In this family, KL-penalty objectives constrain deviations from the old policy, but computing and enforcing KL constraints for large-scale models can be costly. We therefore adopt a clipping-based alternative: token-level clipping directly bounds the likelihood ratio and caps per-token deviations, making it a simple and effective way to constrain updates on off-policy tokens. Concurrently, recent work has also explored this strategy to avoid potential overfitting in SFT (Zhu et al., 2025).

Specifically, the clipping strategy is implemented via a *policy ratio* for a given expert target $y^*$ and input $x$:

$$r(\theta) = \frac{\pi_\theta(y^* \mid x)}{\pi_{\theta_{\text{old}}}(y^* \mid x)}. \quad (5)$$

This ratio quantifies how much the current policy has changed relative to the previous policy for a specific action. To prevent aggressive updates, we moderate supervised learning by clipping this ratio to the bounded interval $[1 - \epsilon, 1 + \epsilon]$, where the hyperparameter $\epsilon$ controls the size of the trust region. Therefore, we set the advantage in Eq. 3 to a positive constant, $\hat{A} = 1$, assigning equal weight to all target behaviors. Thus, we term the resulting variant

*Table 1.* Performance on **ScienceWorld** and **ALFWorld** after cold-start. Metric is success rate(%). Best numbers are bolded.

| Backbone | Method | ScienceWorld | | ALFWorld | | Average |
|---|---|---|---|---|---|---|
| | | Seen | Unseen | Seen | Unseen | |
| **Qwen2.5-7B-Instruct** | ● SFT | 55.15 | 48.34 | 78.57 | 74.63 | 64.17 |
| | ● DFT | 57.73 | 51.18 | 75.71 | **79.85** | 66.12 |
| | ● NEFT | **58.76** | 50.24 | 73.57 | 74.63 | 64.30 |
| | ● OPC-SFT | 58.25 | **54.98** | **82.86** | 78.36 | **68.61** |
| **Qwen2.5-1.5B-Instruct** | ● SFT | 54.12 | 53.08 | 70.71 | 70.90 | 62.20 |
| | ● DFT | 64.43 | 56.40 | 61.43 | 70.90 | 63.29 |
| | ● NEFT | 60.82 | **58.77** | 62.14 | 69.40 | 62.78 |
| | ● OPC-SFT | **65.98** | 58.29 | **72.86** | **72.39** | **67.38** |
| **Llama3.2-3B-Instruct** | ● SFT | 56.70 | 53.55 | 75.00 | 70.90 | 64.04 |
| | ● DFT | **65.98** | 55.92 | 72.14 | 73.88 | 66.98 |
| | ● NEFT | 62.37 | 54.03 | **77.14** | 68.66 | 63.94 |
| | ● OPC-SFT | **65.98** | **64.93** | 76.43 | **77.61** | **71.24** |

*Table 2.* OOD performance under the **ALFWorld** setting. Methods: Base, SFT, DFT, NEFT, and **OPC-SFT**. Metrics are accuracy (%) and pass@1.

| Backbone | Method | MBPP | MMLU | HumanEval | GPQA | LiveCodeBench | MATH500 |
|---|---|---|---|---|---|---|---|
| **Qwen2.5-7B-Instruct** | Base | 79.68 | 71.00 | 73.03 | 33.84 | 61.54 | 76.80 |
| | SFT | 71.69 | 66.30 | 70.35 | 31.31 | 58.91 | 68.40 |
| | DFT | 75.13 | 70.20 | 74.88 | **34.85** | 60.02 | 69.00 |
| | NEFT | 72.22 | 70.10 | 43.90 | 28.79 | 63.01 | 69.60 |
| | **OPC-SFT** | **78.84** | 70.60 | **75.07** | 34.34 | **67.69** | 72.40 |
| **Qwen2.5-1.5B-Instruct** | Base | 58.73 | 60.08 | 69.63 | 21.72 | 14.50 | 52.60 |
| | SFT | 42.60 | 58.68 | 43.60 | 28.79 | 21.70 | 24.00 |
| | DFT | 45.50 | 58.63 | 44.40 | 9.60 | 24.52 | 19.20 |
| | NEFT | 46.31 | **59.17** | **45.01** | 28.52 | 27.04 | 18.80 |
| | **OPC-SFT** | **46.56** | 58.85 | 44.96 | **33.84** | **32.81** | **24.20** |
| **Llama3.2-3B-Instruct** | Base | 57.94 | 62.29 | 38.61 | 27.40 | 33.85 | 35.20 |
| | SFT | 56.61 | 58.47 | 44.39 | 10.61 | 34.50 | 21.20 |
| | DFT | 58.26 | 58.69 | 45.84 | 16.67 | 36.22 | 34.20 |
| | NEFT | 52.18 | 55.95 | 46.03 | 16.67 | 39.73 | 29.80 |
| | **OPC-SFT** | **58.71** | **59.97** | **48.13** | **18.18** | **42.60** | **37.60** |

Off-Policy token Clipped SFT (OPC-SFT), and the loss is:

$$\mathcal{L}_{\text{OPC-SFT}}(\theta) = \\ -\mathbb{E}_{(x,y^*)\sim\mathcal{D}}\left[\min\left(r(\theta),\ \text{clip}(r(\theta), 1-\epsilon, 1+\epsilon)\right)\right]. \quad (6)$$

This loss ensures bounded policy updates. By adapting the already proven clipping strategy, it can constrain the SFT update's deviation from a periodically updated old policy, thereby stabilizing the training process. Specifically, if $r(\theta) > 1 + \epsilon$, indicating that the current policy overemphasizes $y^*$ in the expert dataset, clipping prevents an overly aggressive update, which may lead to forgetting. Furthermore, periodically refreshing the old policy parameters $\theta_{\text{old}}$ allows this trust region to adapt as the model learns, balancing the acquisition of new, specialized knowledge with the retention of general capabilities. Consequently, OPC-SFT mitigates the destructive updates and preserves the model's prior knowledge, leading to more robust generalization.

# 4. Experiments

We conduct a suite of experiments and find that OPC-SFT produces a robust cold-start policy that improves subsequent reinforcement learning performance compared to standard SFT and other strong baselines. Our evaluation primarily focuses on LLM agentic environments. First, in Sec. 4.1.1, we evaluate in-domain generalization by testing adaptation to both seen and unseen task variations. Second, in Sec. 4.1.2, we measure out-of-distribution (OOD) performance on general reasoning tasks, including code generation, math reasoning, and commonsense question answering. Strong OOD results indicate that clipping better preserves prior knowledge, which we expect to improve downstream RL performance (Sec. 4.1.3). Third, to understand the mechanisms driving these performance gains, we conduct diagnostic analyses in Sec. 4.2 by visualizing the model's internal representations via PCA and tracking token probability progression. In Sec. 4.3, we show that OPC-SFT yields smaller gains on math reasoning because the math data is less off-policy for

*Table 3.* Final performance on **ScienceWorld** and **ALFWorld** after RL. Metric is success rate(%). Best numbers are bolded.

| Backbone | Method | ScienceWorld | | ALFWorld | | Average |
| | | Seen | Unseen | Seen | Unseen | |
|---|---|---|---|---|---|---|
| **Qwen2.5-7B-Instruct** | Base + GRPO | 41.75 | 47.87 | 62.86 | 58.96 | 52.86 |
| | SFT + GRPO | 60.82 | 60.19 | 85.00 | 76.87 | 69.97 |
| | DFT + GRPO | 61.34 | 59.24 | 90.71 | 80.06 | 72.84 |
| | NEFT + GRPO | 62.89 | 60.66 | 72.14 | 59.70 | 63.85 |
| | **OPC-SFT + GRPO** | **66.49** | **61.14** | **92.14** | **91.04** | **77.70** |
| **Qwen2.5-1.5B-Instruct** | Base + GRPO | 40.72 | 31.75 | 29.29 | 38.06 | 34.96 |
| | SFT + GRPO | 65.98 | 65.40 | 82.86 | 82.84 | 74.27 |
| | DFT + GRPO | **67.53** | 63.98 | 85.07 | 80.71 | 74.32 |
| | NEFT + GRPO | **67.53** | 63.51 | 82.86 | 67.16 | 70.27 |
| | **OPC-SFT + GRPO** | 65.46 | **68.72** | **90.00** | **94.03** | **79.55** |
| **Llama3.2-3B-Instruct** | Base + GRPO | 44.85 | 44.08 | 0.00 | 0.00 | 22.23 |
| | SFT + GRPO | 70.62 | 63.03 | 92.86 | 89.55 | 79.02 |
| | DFT + GRPO | 67.53 | 64.45 | 81.43 | 76.87 | 72.57 |
| | NEFT + GRPO | 61.86 | 61.86 | 67.86 | 45.52 | 59.28 |
| | **OPC-SFT + GRPO** | **73.71** | **65.40** | **94.29** | **92.54** | **81.49** |

the base model than the agentic-task data. In Sec. 4.4, we perform ablation studies on the clipping ratio $\epsilon$ and old policy update frequency to assess the robustness of OPC-SFT.

**Experimental Setup.** Besides standard SFT, we also evaluate OPC-SFT against two baselines designed to improve SFT robustness. The first is a concurrent work DFT (Wu et al., 2025b) that rescales the SFT objective with the token probability. The second is NEFTune (Jain et al., 2023), a recent regularization technique that adds noise to embedding vectors during training. Our primary evaluation is conducted on the embodied agent environments of ALFWorld (Shridhar et al., 2021) and ScienceWorld (Wang et al., 2022). All models are trained and evaluated on a compute infrastructure equipped with accelerators capable of approximately 312 TFLOPS of BFloat16 (BF16) performance. We select three models for evaluation, including Qwen2.5-7B-Instruct, Qwen2.5-1.5B-Instruct (Qwen et al., 2025) and Llama3.2-3B-Instruct (Grattafiori et al., 2024).

### 4.1. Agentic Cold-Start

Agentic tasks provide an ideal testbed for OPC-SFT, as succeeding in these environments requires the LLM to adopt a strict action format (Yao et al., 2024) that is often highly off-policy for a general-purpose model. Deviations from this format such as generating semantically vague instructions like 'move somewhere' or syntactically invalid commands, which can cause execution errors, terminate the environmental interaction, and lead to unpredictable agent behavior. This acute sensitivity to format compliance makes a robust SFT cold-start essential for successfully initializing the LLM policy, equipping it with the required action format so that it can generalize beyond the off-policy expert data.

#### 4.1.1. IN-DISTRIBUTION VALIDATION BEFORE RL

We evaluate OPC-SFT on two agentic benchmarks: ALF-World (Shridhar et al., 2021) and ScienceWorld (Wang et al., 2022). A key advantage of these tasks is their setup for evaluating generalization within the target domain. They provide test sets with unseen instances that require the model to apply its learned knowledge to new scenarios that are variants of tasks during training. Specifically, the ALFWorld benchmark is composed of 140 seen and 134 unseen test samples, while ScienceWorld contains 194 seen and 211 unseen samples. To ensure a fair comparison, we adhere to standard evaluation protocols from EMBod-Bench (Fei et al., 2025) for ALFWorld and ScienceWorld. As shown in Tab. 1, OPC-SFT achieves comparable performance against all the compelling methods.

#### 4.1.2. OUT-OF-DISTRIBUTION VALIDATION BEFORE RL

To evaluate knowledge retention and OOD generalization, we test the fine-tuned models on a suite of standard benchmarks. These include coding tasks, like MBPP (Austin et al., 2021) and HumanEval (Chen et al., 2021), general knowledge assessments, like MMLU (Hendrycks et al., 2021a) and GPQA (Rein et al., 2023), and mathematical reasoning MATH-500 (Hendrycks et al., 2021b). For GPQA, we use the GPQA Diamond subset. The results in Tab. 2 and Tab. 7 (deferred to Appx. B.2) highlight the 'alignment tax' of standard SFT, which exhibits significant performance degradation. OPC-SFT better preserves prior knowledge, reducing the average performance drop by 11.54% relative to standard SFT. Notably, Qwen2.5-7B-Instruct trained with OPC-SFT achieves the strongest anti-forgetting performance among all baselines. Preserving general capabilities from pre-training like commonsense and logical reasoning

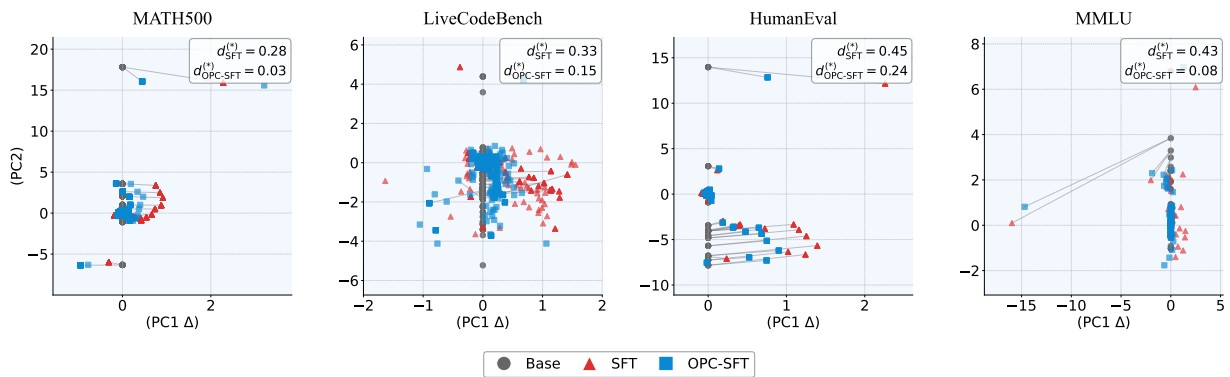

*Figure 3.* PCA shift of Llama3.2-3B-Instruct with the SFT and OPC-SFT methods.

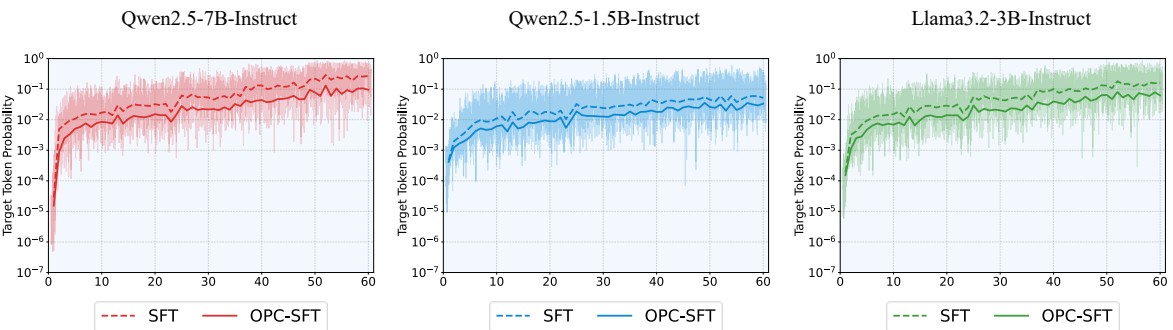

*Figure 4.* Token probability in the bottom 1% quantile over training steps for SFT and OPC-SFT.

supports downstream agentic exploration and generalization (Zhao et al., 2025). Since a high alignment tax limits downstream RL adaptation, which is shown in Sec. 4.1.3, reducing this tax makes OPC-SFT a stronger cold-start for subsequent reinforcement learning.

### 4.1.3. PERFORMANCE COMPARISON AFTER RL

A robust cold-start is important, but the ultimate test is the performance after RL (DeepSeek-AI et al., 2025). We therefore initialize policies with each method and further train them using Group Relative Policy Optimization (GRPO) (Shao et al., 2024). As shown in Tab. 3, OPC-SFT yields the strongest downstream RL performance, with particularly large gains on ALFWorld. In contrast, applying GRPO directly to the base model performs poorly, underscoring the need for an effective cold-start in agentic tasks.

### 4.2. Latent Space Shift Analysis and Target Token Probability Change over Training Steps

We conduct internal representation and gradient analysis to account for the advantages of OPC-SFT. Xu et al. (2025) shows that PCA shift analysis serves as a sensitive and interpretable metric for representational changes associated with task performance. We decompose the latent space of the LLM on the target domain using PCA ($n = 2$) projection.

We can observe from Fig. 3 that OPC-SFT exhibits a smaller divergence from the base model compared to SFT.

Specifically, when evaluated by Euclidean distance $d^{(*)} = \|\mathbf{z}^{(*)} - \mathbf{z}^{(\text{orig})}\|_2$, where $\mathbf{z}$ denotes the mean PCA coordinates of hidden states across layers in the low-dimensional space. SFT yields divergences of 0.28, 0.33, 0.45, and 0.43 on the MATH500, LiveCodeBench, HumanEval, and MMLU benchmarks, respectively. In contrast, OPC-SFT significantly reduces these divergences, yielding 0.03, 0.15, 0.24, and 0.08 on the same respective benchmarks. The projection details are deferred to Appx. C.1. Furthermore, analysis of the target token probability progression, as seen in Fig. 4, reveals that OPC-SFT increases target token probabilities more steadily. Clipping low-probability tokens enables the model to fit new targets while minimizing the loss of prior knowledge.

### 4.3. Experiments on Math Reasoning Task

To investigate mathematical reasoning performance, we fine-tune the LLM on data collected from the DeepScaleR (Luo et al., 2025b) problem suite and report the final performance in Appx. B.1 (Tab. 4). While OPC-SFT still outperforms the baseline, the performance gains are more modest than those in the agentic tasks. This finding, which is consistent with concurrent work (Zhu et al., 2025), prompts us to

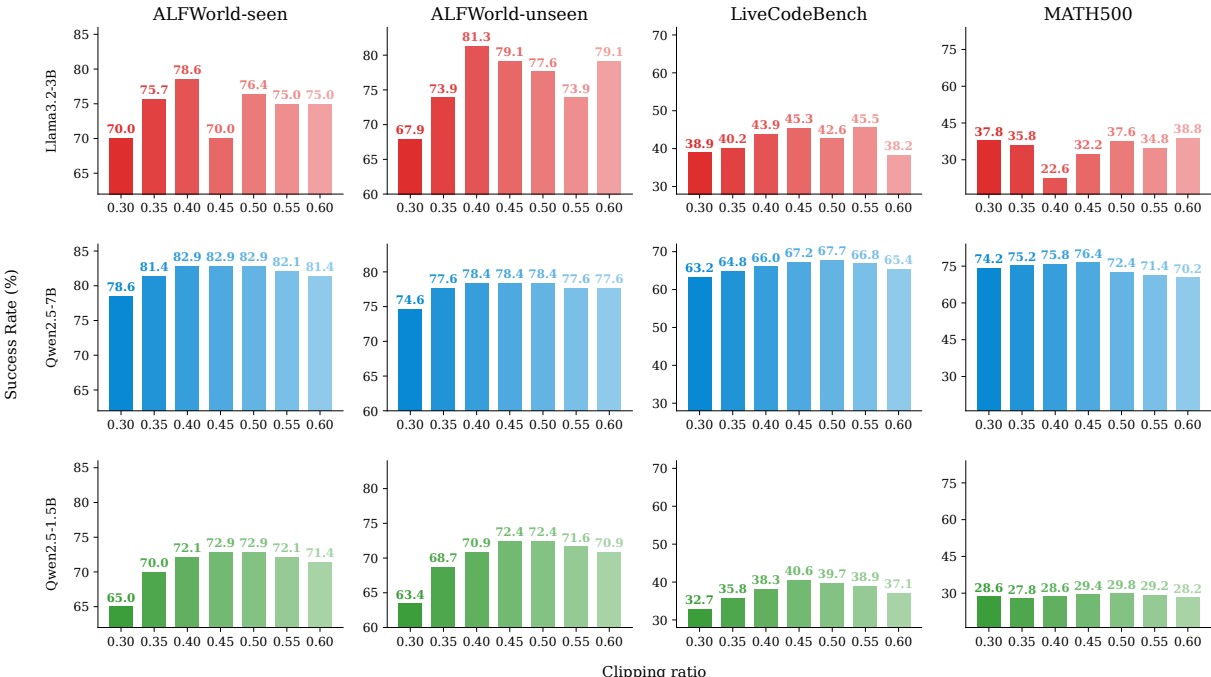

*Figure 5.* Clipping ratio ablation on Llama3.2-3B-Instruct on ALFWorld before RL.

investigate the conditions under which OPC-SFT is most effective. We hypothesize that the performance discrepancy is due to the nature of the fine-tuning data. Unlike the novel action formats in agentic tasks, mathematical reasoning is already well-represented in the LLMs' pre-training corpora. Consequently, SFT for math involves a smaller distributional shift, resulting in less of the off-policy instability that OPC-SFT is designed for. To test this hypothesis, we analyze the gradient norm distributions at the beginning of the SFT phase for both task types. The result in Appx. B.1 (Fig. 7) shows that the gradient norms for the agentic task data are substantially larger than those for the math reasoning data. This evidence demonstrates that the benefits of OPC-SFT are pronounced when the fine-tuning data is off-policy.

### 4.4. Ablation Studies

**On clipping ratio.** We conduct an ablation study to understand the influence of the clipping ratio $\epsilon$ on downstream performance. Across three model backbones, we vary $\epsilon$ while fixing the old policy update interval to 256 samples. As shown in Fig. 5, performance is generally stable for a moderate range of clipping ratios, while the best value is not identical across tasks and models. For example, on Llama3.2-3B-Instruct, smaller ratios perform well on ALFWorld, whereas larger ratios can be better on MATH500 but may hurt LiveCodeBench. Overall, $\epsilon = 0.50$ provides a robust trade-off across benchmarks and backbones, and we use it as the default setting.

**On old policy update frequency.** A critical component of

our optimization framework is the old policy model $\pi_{\theta_{\text{old}}}$. As training progresses, the current policy $\pi_\theta$ naturally diverges from earlier states. Hence, we conduct an ablation study on the update frequency, defined as the number of samples used to train the policy model before resetting the old policy. We evaluate Llama3.2-3B-Instruct using intervals of 128, 256, 512, and 1024 samples. Fig. 6 and Tab. 8 in Appx. B.3 show that larger intervals lead to substantial degradation, confirming that a 'stale' old policy over-constrains the optimization and prevents the model from adopting improved behaviors. Conversely, updating too frequently yields slightly worse performance, likely because the trust region shifts too rapidly to provide a stable old policy.

**Isolating Clipping from implicit data reweighting.** The objective form in OPC-SFT couples two effects: it implicitly reweights tokens through the ratio, and it bounds large updates through clipping. To isolate the source of gains, we perform a controlled ablation that removes clipping while keeping the same ratio-based reweighting. As shown in Appx. B.3 (Fig. 8), ratio-based reweighting alone yields modest improvements on in-distribution metrics, but provides limited gains for out-of-distribution generalization.

### 5. Conclusion

SFT cold-start yields the initial policy for reinforcement learning in LLMs. For efficient downstream RL, this initial model must strike a delicate balance: acquiring specialized behaviors while retaining broad prior knowledge. In this

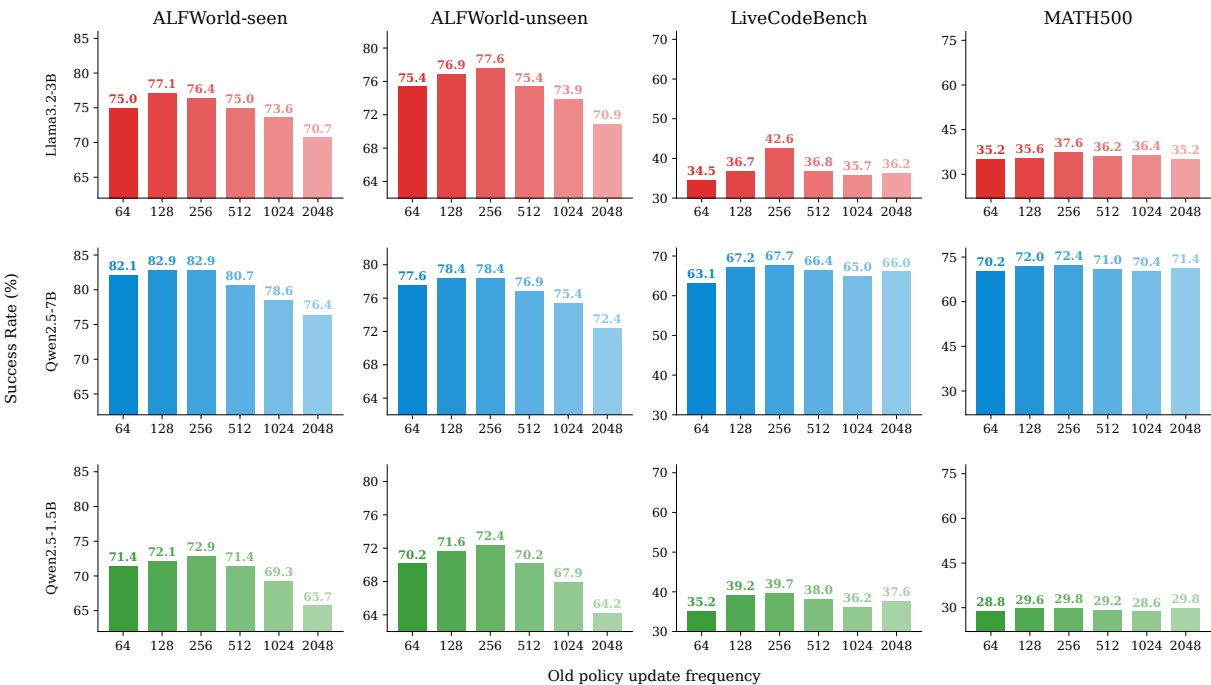

*Figure 6.* Old policy update frequency ablation on Llama3.2-3B-Instruct on ALFWorld before RL.

work, we identify a key failure mode of the SFT cold-start: degraded generalization and loss of pre-trained knowledge. Our analyses further indicate that this forgetting is closely tied to large gradient norms early in fine-tuning, which are dominated by a small subset of low-probability tokens. Motivated by this diagnosis, we evaluate a simple token-level clipping intervention, OPC-SFT, that bounds the updates induced by off-policy tokens. Across agentic benchmarks, OPC-SFT consistently improves out-of-distribution generalization and reduces forgetting, yielding a more robust cold-start for subsequent RL. Further analyses show that the clipping strategy is a practical way to mitigate forgetting. While OPC-SFT is effective, it has limitations. If clipping is set too tightly, it can overly restrict learning from off-policy tokens and shift the learning burden to medium-probability tokens, which may destabilize optimization and, in extreme cases, reduce model entropy.

## Acknowledgements

This work was supported by the National Natural Science Foundation of China under Grants 62495090, 62495093, and the Natural Science Foundation of Jiangsu under Grants BK20243039, and the "111 Center" (No. B26023), Fundamental and Interdisciplinary Disciplines Breakthrough Plan of the Ministry of Education of China (No. JYB2025XDXM118).

## Impact Statement

This paper presents work whose goal is to advance the field of machine learning. There are many potential societal consequences of our work, none of which we feel must be specifically highlighted here.

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

# A. Explanatory Analysis

## A.1. Clipping Strategy Details

We present the analysis of the clipping mechanisms in OPC-SFT. To ground this analysis, we define the optimization context on a supervised learning dataset $\mathcal{D} = \{(x, y^*)\}$. Drawing inspiration from the well-established clipping strategy in PPO (Schulman et al., 2017), we implement a token-level trust region to explicitly manage off-policy tokens and prevent destructive policy updates.

The gradient of the OPC-SFT loss inherits the stabilizing behavior of the trust region method:

$$\nabla_\theta \mathcal{L}_{\text{OPC-SFT}}(\theta) = -\mathbb{E}_{(x,y^*)\sim\mathcal{D}} \left[ \nabla_\theta \left( \min \left( r(\theta), \text{clip}(r(\theta), 1 - \epsilon, 1 + \epsilon) \right) \right) \right].$$

Here, $r(\theta)$ represents the probability ratio for a specific token. This gradient behaves in two distinct cases:

When $r(\theta) \leq 1 + \epsilon$, the gradient uses $r(\theta)$, enabling meaningful learning:

$$\nabla_\theta r(\theta) = r(\theta) \cdot \nabla_\theta \log \pi_\theta(y^* \mid x).$$

When $r(\theta) > 1 + \epsilon$, the clipped term dominates the $\min$, so the gradient is zero:

$$\nabla_\theta \left( \text{clip}(r(\theta), 1 - \epsilon, 1 + \epsilon) \right) = 0.$$

By strictly enforcing this token-level constraint, OPC-SFT suppresses gradients for samples where the policy deviates too far from the old policy, preventing the aggressive updates often associated with off-policy tokens.

## A.2. Clipping as Margin Control

We show that assigning high confidence to a target token with very low reference probability requires a large margin correction. OPC-SFT bounds the probability ratio within each learning interval, and also bounds the corresponding margin increase, thereby reducing representational drift and better preserving the LLM's pre-trained generalization ability.

Let $m_\theta(x, y^*) = z_\theta(x, y^*) - \log \sum_{y \neq y^*} \exp(z_\theta(x, y))$ denote the target-token margin. Since $\pi_\theta(y^* \mid x) = \sigma(m_\theta(x, y^*))$, increasing the target-token probability is equivalent to increasing its margin.

Suppose $\delta := \pi_{\theta_{\text{old}}}(y^* \mid x) \ll 1$, and we want $\pi_\theta(y^* \mid x) \geq 1 - \xi$ for some fixed $\xi \in (0, 1)$. Then the required margin increase satisfies

$$\Delta m_{\text{target}} \geq \log \frac{1 - \xi}{\xi} - \log \frac{\delta}{1 - \delta} = \Theta\left(\log \frac{1}{\delta}\right) \qquad (\delta \ll 1).$$

Now assume the margin is locally smooth in parameter space: $\|\nabla_\theta m_\theta(x, y^*)\|_2 \leq C$ along the relevant optimization path. For any continuous path $\gamma : [0, 1] \to \mathbb{R}^d$ from $\theta_{\text{old}}$ to $\theta$, the fundamental theorem of calculus gives

$$m_\theta(x, y^*) - m_{\theta_{\text{old}}}(x, y^*) = \int_0^1 \langle \nabla_\theta m_{\gamma(t)}(x, y^*), \gamma'(t) \rangle dt.$$

Applying Cauchy-Schwarz,

$$\Delta m_{\text{target}} \leq C \int_0^1 \|\gamma'(t)\|_2 \, dt.$$

Defining the path length by $\text{Len}(\gamma) := \int_0^1 \|\gamma'(t)\|_2 \, dt$,

we obtain

$$\text{Len}(\gamma) \geq \frac{1}{C} \Delta m_{\text{target}} = \Omega\left(\frac{1}{C} \log \frac{1}{\delta}\right).$$

By contrast, within a fixed reference interval of OPC-SFT, clipping enforces $\frac{\pi_\theta(y^*|x)}{\pi_{\theta_{\text{old}}}(y^*|x)} \leq 1 + \epsilon$. Since $\pi_{\theta_{\text{old}}}(y^* \mid x) = \delta$, this implies

$$\pi_\theta(y^* \mid x) \leq (1 + \epsilon)\delta.$$

Assuming $(1 + \epsilon)\delta < 1$, this yields

$$m_\theta(x, y^*) \leq \log \frac{(1 + \epsilon)\delta}{1 - (1 + \epsilon)\delta},$$

and hence the margin increase within one interval is bounded by

$$\Delta m_{\text{OPC-SFT,interval}} \leq \log \frac{(1 + \epsilon)\delta}{1 - (1 + \epsilon)\delta} - \log \frac{\delta}{1 - \delta}.$$

For $\delta \ll 1$, we get

$$\Delta m_{\text{OPC-SFT,interval}} = \log(1 + \epsilon) + O(\delta\epsilon) = O(\log(1 + \epsilon)).$$

Therefore, while reaching high confidence may require a total correction of order $\Theta(\log(1/\delta))$, OPC-SFT restricts the correction within each reference interval to only $O(\log(1 + \epsilon))$. Finally, if the hidden representation $h_\theta^{(l)}(x)$ is locally Lipschitz with respect to the parameters, then representational change is controlled by parameter movement. Hence, bounding the per-interval correction scale also gives a tighter bound on abrupt representational drift.

## B. Additional Experiments

### B.1. OPC-SFT on Math Reasoning Tasks

*Table 4.* Final mathematical reasoning performance after the RL phase. LLMs initialized with different cold-start methods are trained with GRPO. Metrics are accuracy (%) and avg@8.

| Backbone | Method | Minerva | Olympiad Bench | GSM8K | AIME24 | MATH500 | Average |
|---|---|---|---|---|---|---|---|
| Qwen2.5-7B-Instruct | SFT + *GRPO* | 39.71 | 47.45 | 94.31 | 21.67 | 84.00 | 57.43 |
| | NEFT + *GRPO* | 39.78 | 47.52 | 94.38 | 21.75 | 84.05 | 57.50 |
| | DFT + *GRPO* | **39.82** | 47.58 | 94.42 | 21.83 | 84.08 | 57.55 |
| | PSFT$_{warm\,up}$ + *GRPO* | 39.04 | 46.82 | 93.84 | 19.00 | 83.20 | 56.38 |
| | **OPC-SFT + *GRPO*** | 39.34 | **48.33** | **95.68** | **23.33** | **84.35** | **58.21** |
| Qwen2.5-1.5B-Instruct | SFT + *GRPO* | 30.52 | 38.56 | 84.95 | **17.50** | 77.30 | 49.77 |
| | NEFT + *GRPO* | 30.28 | 38.41 | 84.75 | 16.83 | 77.02 | 49.46 |
| | DFT + *GRPO* | **30.85** | **40.84** | **86.13** | 17.05 | **77.52** | **50.48** |
| | PSFT$_{warm\,up}$ + *GRPO* | 29.68 | 38.02 | 84.51 | 16.25 | 76.70 | 49.03 |
| | **OPC-SFT + *GRPO*** | 29.81 | 40.74 | 85.97 | 15.41 | 77.20 | 49.83 |
| Llama3.2-3B-Instruct | SFT + *GRPO* | 19.71 | 22.08 | 79.88 | **9.58** | 56.60 | 37.57 |
| | NEFT + *GRPO* | **19.78** | 22.12 | 79.95 | 9.38 | 56.48 | 37.54 |
| | DFT + *GRPO* | 19.75 | 22.13 | 79.97 | 9.41 | 56.55 | 37.56 |
| | PSFT$_{warm\,up}$ + *GRPO* | 19.12 | 21.64 | 79.45 | 8.87 | 56.00 | 37.02 |
| | **OPC-SFT + *GRPO*** | 20.59 | **22.51** | **80.54** | 9.55 | **57.10** | **37.86** |

We show the performance of OPC-SFT and SFT on mathematical reasoning benchmarks, as shown in Tab. 4, Tab. 5 and their generalization to common-sense question-answering MMLU (Hendrycks et al., 2021a), GPQA (Rein et al., 2023), and coding tasks HumanEval (Chen et al., 2021) and LiveCodeBench (Jain et al., 2025), as shown in Tab. 6. These results show that the benefits of OPC-SFT are most pronounced when the expert data is highly off-policy. This scenario could result in large gradient norms at the initial stage of cold-start, which is validated in Fig. 7.

### B.2. Missing OOD Evaluation

Due to the strict page limit, we present the OOD performance for models trained on ScienceWorld in Tab. 7. Across the majority of these tasks, OPC-SFT either outperforms all other baselines or achieves the second-best result. These findings,

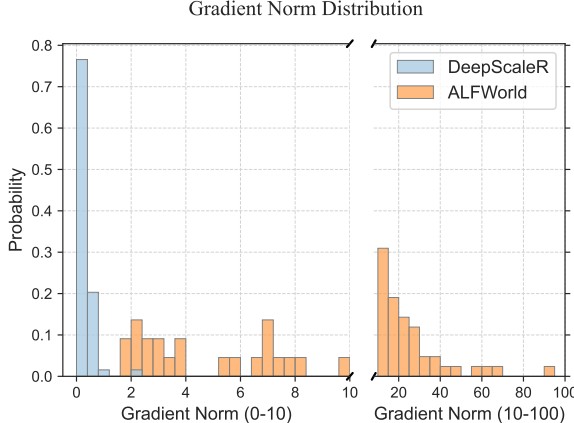

*Figure 7.* Gradient norm comparison between math task DeepScaleR and agentic task ALFWorld.

*Table 5.* Mathematical reasoning performance of the cold-start policies (before RL). OPC-SFT is compared against the standard SFT baseline. Metrics are accuracy (%) and avg@8.

| Backbone | Method | Minerva | Olympiad Bench | GSM8K | AIME24 | MATH500 | Average |
|---|---|---|---|---|---|---|---|
| Qwen2.5-7B-Instruct | Base | 38.24 | 36.37 | 91.89 | 13.75 | 76.80 | 51.41 |
| | SFT | 39.04 | 42.37 | **94.24** | **18.33** | 81.80 | 55.16 |
| | OPC-SFT | **41.48** | **43.56** | 93.78 | 15.83 | 81.40 | **55.21** |
| Qwen2.5-1.5B-Instruct | Base | 13.76 | 18.07 | 70.58 | 1.67 | 52.60 | 31.34 |
| | SFT | 24.26 | 36.89 | 82.11 | **10.86** | 74.40 | 45.70 |
| | OPC-SFT | **26.16** | **38.11** | **84.91** | 10.00 | **76.60** | **47.16** |
| Llama3.2-3B-Instruct | Base | 10.56 | 9.63 | 67.25 | 0.83 | 35.20 | 24.69 |
| | SFT | 14.31 | 18.07 | 76.63 | 1.67 | **52.20** | 32.58 |
| | OPC-SFT | **16.54** | **18.34** | **77.15** | **3.34** | 51.80 | **33.43** |

*Table 6.* OOD performance of models for different cold-start. OPC-SFT + GRPO is compared against SFT + GRPO baseline. Metrics are accuracy (%) and pass@1.

| Backbone | Method | MMLU | HumanEval | GPQA | LiveCodeBench | Average |
|---|---|---|---|---|---|---|
| Qwen2.5-7B-Instruct | Base | 71.00 | 73.03 | 33.84 | 61.54 | 59.85 |
| | SFT | 64.43 | 1.07 | 35.35 | 0.20 | 25.26 |
| | OPC-SFT | **66.64** | **2.74** | **35.35** | **0.31** | **26.26** |
| Qwen2.5-1.5B-Instruct | Base | 60.08 | 69.63 | 21.72 | 14.50 | 41.48 |
| | SFT | 58.31 | 0.00 | 27.40 | 0.00 | **21.43** |
| | OPC-SFT | **58.62** | 0.00 | 26.77 | 0.00 | 21.35 |
| Llama3.2-3B-Instruct | Base | 62.29 | 38.61 | 27.78 | 33.85 | 40.63 |
| | SFT | 54.90 | 16.84 | 30.30 | 0.00 | 25.51 |
| | OPC-SFT | **55.45** | **17.37** | **31.31** | 0.00 | **26.03** |

*Table 7.* OOD performance under the **ScienceWorld** setting. Methods: Base, SFT, DFT, NEFT, and OPC-SFT. Metrics are accuracy (%) and pass@1.

| Backbone | Method | MBPP | MMLU | HumanEval | GPQA | LiveCodeBench | MATH500 |
|----------|--------|------|------|-----------|------|---------------|---------|
| **Qwen2.5-7B-Instruct** | Base | 79.68 | 71.00 | 73.03 | 33.84 | 61.54 | 76.80 |
| | SFT | 67.20 | 64.43 | 64.63 | 46.46 | 49.55 | 56.40 |
| | DFT | 69.33 | 54.35 | 62.71 | 41.92 | 52.71 | 55.20 |
| | NEFT | 69.58 | **67.03** | 63.53 | 37.88 | 50.09 | 46.80 |
| | **OPC-SFT** | **71.96** | 66.64 | **69.13** | **63.64** | 55.59 | **57.40** |
| **Qwen2.5-1.5B-Instruct** | Base | 58.73 | 60.08 | 69.63 | 21.72 | 14.50 | 52.60 |
| | SFT | **49.74** | 58.31 | 41.28 | 30.81 | 7.73 | 42.80 |
| | DFT | 46.03 | 57.47 | 46.32 | **51.52** | 9.66 | 35.20 |
| | NEFT | 45.02 | 57.68 | 44.44 | 46.46 | 9.52 | 39.80 |
| | **OPC-SFT** | 48.15 | **58.62** | **47.82** | 50.51 | **11.83** | **43.40** |
| **Llama3.2-3B-Instruct** | Base | 57.94 | 62.29 | 38.61 | 27.78 | 33.85 | 35.20 |
| | SFT | 54.50 | 54.90 | 37.19 | 40.40 | 23.07 | 29.00 |
| | DFT | **58.11** | 55.95 | **39.12** | 50.00 | 24.59 | 32.80 |
| | NEFT | 55.03 | **57.19** | 32.31 | 45.96 | 25.15 | 31.20 |
| | **OPC-SFT** | 56.35 | 55.45 | 38.49 | **51.01** | **27.39** | **34.20** |

together with the results from ALFWorld, as shown in Tab. 2, demonstrate that OPC-SFT effectively retains the model's generalizable prior knowledge.

### B.3. Missing Ablation Study Result

Due to the strict page limit, we present the missing ablation results here. Tab. 8, along with Fig. 6 in the main text, shows that OPC-SFT is sensitive to the update frequency of the old policy: if $\pi_{\theta_{old}}$ is updated too infrequently, the ratio becomes increasingly stale, and clipping becomes overly restrictive, which can hinder learning on the expert data by clipping more tokens than intended. Fig. 8 further isolates clipping from the implicit ratio-based reweighting. Reweighting alone yields only marginal gains on in-distribution evaluations, but leads to substantially worse OOD performance, indicating that it accelerates fitting to the target task without mitigating forgetting.

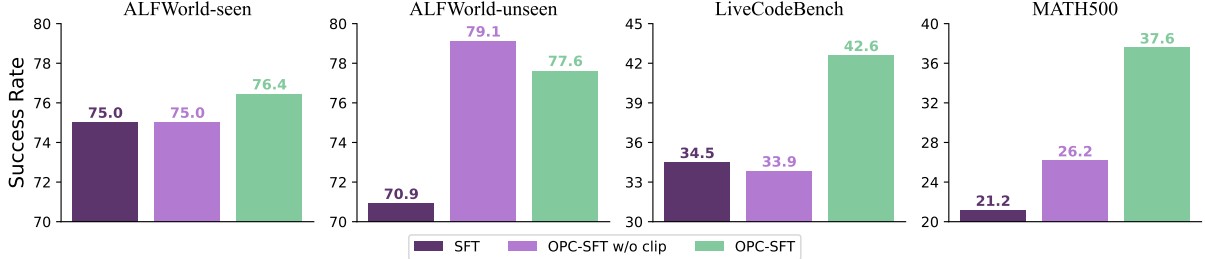

*Figure 8.* A controlled study of the clip component in OPC-SFT on Llama-3.2-3B-Instruct for ALFWorld.

### B.4. Agentic Task with AgentBoard Validation Dataset

Besides EMBod-Bench (Fei et al., 2025) used in Sec. 4.1.1, we also adopt the AgentBoard (Ma et al., 2024) framework to conduct a two-stage experiment: first performing SFT, followed by RL, aiming to investigate the cold-start performance of different SFT approaches. AgentBoard is a benchmark for evaluating multi-turn LLM agents. It spans nine task categories and over a thousand environments, encompassing widely used benchmarks such as ALFWorld and ScienceWorld, which capture multi-round and partially observable settings. Its accompanying open-source toolkit further enables detailed analysis through visualization of trajectories, skill-specific performance, and difficulty breakdowns, providing a comprehensive diagnostic framework for agent research.

We conduct experiments on the ALFWorld and ScienceWorld benchmarks. Compared to EMBod-Bench (Fei et al., 2025), the test sets of these two benchmarks in the AgentBoard framework differ as follows: ALFWorld contains 134 unseen test instances that overlap with those in EMBod-Bench, whereas ScienceWorld includes 90 unseen test instances that are distinct from those in EMBod-Bench. In addition, the inference settings of AgentBoard and EMBod-Bench are not identical. The

*Table 8.* OOD performance for the old policy update frequency ablation.

| $\pi_{\theta_{old}}$ Update Freq | MBPP | MMLU | HumanEval | GPQA | LiveCodeBench | MATH500 |
|---|---|---|---|---|---|---|
| 128 | 58.99 | 57.84 | 47.98 | 17.17 | 36.65 | 35.60 |
| 256 | 58.71 | **59.97** | 48.13 | 18.18 | **42.60** | **37.60** |
| 512 | **61.11** | 57.03 | **48.89** | 19.70 | 36.79 | 36.20 |
| 1024 | 60.05 | 59.26 | 46.53 | **21.21** | 35.68 | 36.40 |

*Table 9.* SFT and SFT+RL Performance on **ScienceWorld** and **ALFWorld**. Metrics are success rate (%). Best numbers are bolded.

| Backbone | Method | ScienceWorld | | ALFWorld | |
|---|---|---|---|---|---|
| | | Before RL | After RL | Before RL | After RL |
| **Qwen2.5-7B-Instruct** | SFT | 34.44 | 43.33 | 76.87 | 84.33 |
| | DFT | 37.78 | 44.44 | 75.37 | 80.60 |
| | NEFT | 41.11 | 48.89 | 77.61 | 64.92 |
| | OPC-SFT | **50.00** | **58.89** | **79.10** | **90.29** |
| **Qwen2.5-1.5B-Instruct** | SFT | 32.22 | 45.56 | 73.88 | 82.08 |
| | DFT | 30.00 | 50.00 | 67.91 | 82.83 |
| | NEFT | 37.78 | 53.33 | 73.13 | 82.83 |
| | OPC-SFT | **40.00** | **55.56** | **76.12** | **87.31** |
| **Llama3.2-3B-Instruct** | SFT | 37.78 | 41.11 | 72.39 | 83.58 |
| | DFT | 41.11 | 46.67 | 72.39 | 55.00 |
| | NEFT | 43.33 | 44.44 | 70.90 | 56.71 |
| | OPC-SFT | **53.33** | **61.10** | **76.87** | **91.79** |

results are shown in Tab. 9.

ALFWorld (Shridhar et al., 2021) consists of planning tasks situated in household settings, ranging from basic object manipulation (e.g., pick-and-place) to scenarios that demand multi-step interactions. For instance, in the "discard a card" task, the agent must first identify the target card, pick it up, locate a trash bin, and correctly dispose of the card to complete the task.

ScienceWorld (Wang et al., 2022) is a challenging benchmark that requires models to carry out scientific experiments in an interactive environment. The environment is supported by a physics engine that incorporates thermodynamic and electrical systems, thus demanding strong planning and causal reasoning skills. For example, one task may ask: turn on the red light bulb by powering it using a renewable power source.

### B.5. Target Token Distribution During OPC-SFT

Supervised Fine-Tuning is widely used to enhance the performance of Large Language Models on task-specific objectives. In SFT, the model is trained on a dataset of high-quality input-output pairs, which are typically derived from expert demonstrations or synthetic trajectories generated by teacher models. Through this process, the LLM learns structured reasoning patterns, task-specific knowledge, and preferred action strategies.

To illustrate the effect of OPC-SFT, we analyze the distribution of token-level probabilities at the beginning and at the end of training. We divide the probability, which range from 0 to 1, into discrete intervals and plot the proportion of target tokens falling into each interval. In the initial stage of training, a noticeable fraction of target tokens have low probability, indicating uncertain predictions. By the end of training, the distribution shifts significantly: nearly all target tokens attain higher probability, reflecting increased confidence and better alignment with the expert trajectories as shown in Fig. 9. This visualization quantitatively demonstrates how SFT improves the model's certainty and task-specific performance.

### B.6. Training Reward Curve in Reinforcement Learning

Large Language Models first acquire general reasoning and task-specific patterns through SFT, providing a strong and high-performing initialization for subsequent RL. In this study, after SFT, we further train LLMs in two benchmark environments: ALFWorld and ScienceWorld, to adapt the pre-trained models to interactive, sequential decision-making

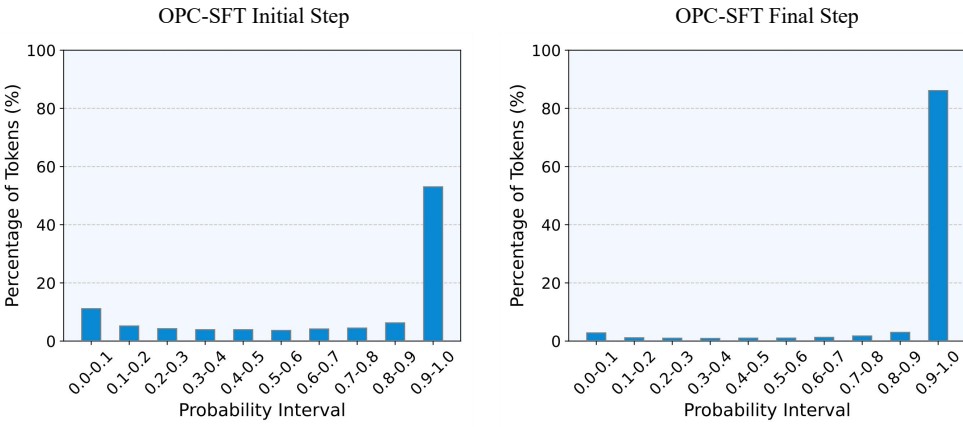

*Figure 9.* Target token probability distribution at the beginning and end of OPC-SFT. The x-axis shows probability intervals, and the y-axis shows the proportion of tokens in each interval.

tasks. This two-stage training paradigm allows the models to start from a higher baseline, which facilitates more effective exploration and accelerates policy refinement through trial-and-error interactions in the environment. Each LLM is trained using one of four strategies: SFT, DFT, NEFT, and OPC-SFT, where OPC-SFT incorporates on-policy correction for cold-start to improve stability and sample efficiency.

Fig. 10 shows the normalized training rewards over the first 100 steps. The top row corresponds to ALFWorld, and the bottom row corresponds to ScienceWorld. Each subplot contains four curves representing the different training strategies: SFT, DFT, NEFT, and OPC-SFT. Here, the reward indicates the success of a trajectory: 1 for success and 0 for failure. Solid lines represent smoothed rewards, while semi-transparent lines show raw values.

From Fig. 10, it is clear that OPC-SFT consistently achieves higher rewards across both environments. Starting from a strong SFT initialization gives OPC-SFT a higher starting point, which not only accelerates early performance but also encourages more effective exploration, enabling the model to discover successful trajectories faster. These results highlight the importance of combining supervised pre-training with on-policy RL correction: the LLMs first acquire structured reasoning and task knowledge via SFT, and then efficiently adapt their policies to maximize task success through RL. Overall, this two-stage approach enables LLMs to leverage prior knowledge while learning interactive behaviors, achieving both sample-efficient learning and robust task performance.

### B.7. The Changes in Clipped Token Counts During Old Policy Updates

In OPC-SFT, the old policy $\pi_{\theta_{\text{old}}}$ is periodically updated, and the percentage of clipped tokens decreases as training progresses. Eventually, the policy will converge, and $\frac{\pi_\theta}{\pi_{\theta_{\text{old}}}}$ will approximate 1, not exceeding $1 + \epsilon$. As a result, the number of clipped tokens decreases as the gradient magnitude gradually becomes smaller until convergence. We provide the corresponding figure for ALFWorld with $\epsilon = 0.5$, as shown in Fig. 11. If the clipping ratio is set to a larger value, the number of clipped tokens will decrease further.

## C. Experimental Details

### C.1. PCA Details

Given a batch of queries $x$, we extract hidden states $H_i^{(*)}(x)$ at each layer $i$ for both model states $(*) \in \{\text{orig}, \text{upd}\}$. Principal Component Analysis (PCA) with $n = 2$ is then applied to $H_i^{(*)}$, and the mean projections onto the first and second principal directions (PC1 and PC2) are denoted by $m_{i,1}^{(*)}$ and $m_{i,2}^{(*)}$, respectively. The shift along PC1 is defined as

$$\Delta m_{i,1}^{(*)} = m_{i,1}^{(*)} - m_{i,1}^{(\text{orig})},$$

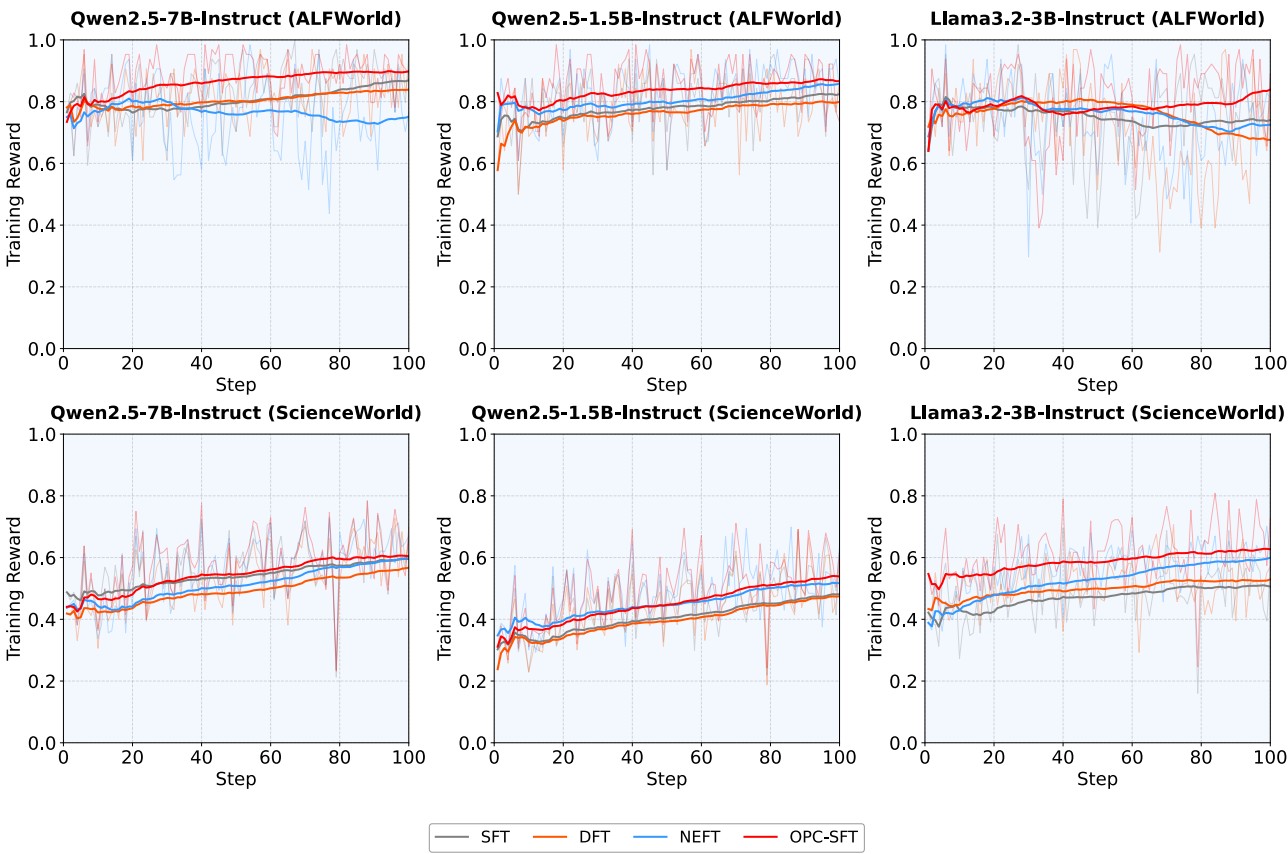

*Figure 10.* Training reward comparison across models and environments. The top row shows results in the ALFWorld environment for three models, and the bottom row shows results in the ScienceWorld environment. Each subplot contains four curves corresponding to SFT, DFT, NEFT, and OPC-SFT. Solid lines represent smoothed reward values calculated as a running average, and semi-transparent lines show the raw rewards recorded during training. The x-axis denotes training steps, and the y-axis denotes the normalized reward.

while $m_{i,2}^{(*)}$ is reported for PC2 as an auxiliary indicator of distributional variation, with smaller values reflecting more stable representations. For each model state $(*)$, we define a representation center as the mean of PCA coordinates across all layers:

$$\mathbf{z}^{(*)} = \frac{1}{L} \sum_{i=1}^{L} \mathbf{z}_i^{(*)}, \quad \mathbf{z}_i^{(*)} = \left(\Delta m_{i,1}^{(*)}, m_{i,2}^{(*)}\right),$$

where $L$ denotes the total number of layers. Finally, the latent shift between state $(*)$ and the original (base) model is quantified by the Euclidean distance

$$d^{(*)} = \|\mathbf{z}^{(*)} - \mathbf{z}^{(\text{orig})}\|_2.$$

## C.2. ALFWorld Demonstration

To concretely illustrate LLM reasoning in embodied settings, we present an example interaction in ALFWorld (Shridhar et al., 2021). ALFWorld is a text-based, interactive household environment derived from ALFRED, where an agent must execute natural-language goals like cleaning, heating or placing objects by issuing grounded, step-wise actions based on partial observations. Solving these tasks requires multi-step planning, state tracking, and compositional reasoning over objects and locations. In our demonstration, prompts are constructed in a multi-turn manner: at each step, we concatenate the current observation with the model's previous output to condition the next prediction, mirroring the agent's online decision-making loop. The dialogue below highlights this iterative interaction process, showing how the LLM produces structured outputs and updates its behavior as new observations arrive.

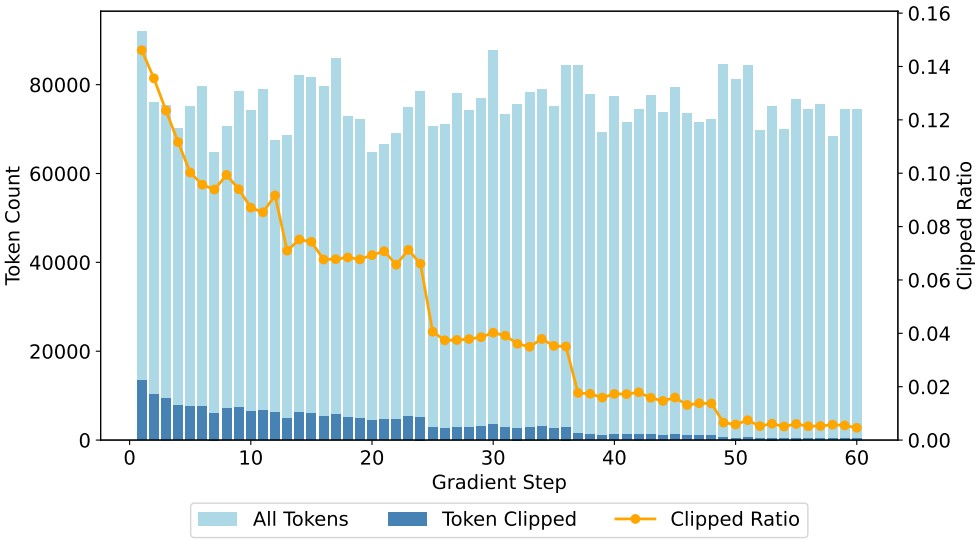

*Figure 11.* Changes in clipped tokens over steps when using OPC-SFT with $\epsilon = 0.5$ on ALFWorld

---

**System**

You are an expert in the ALFRED Embodied Environment.

---

**User**

Interact with a household to solve a task. Imagine you are an intelligent agent in a household environment and your target is to perform actions to complete the task goal. At the beginning of your interactions, you will be given the detailed description of the current environment and your goal to accomplish.

For each of your turn, you will be given the observation of the last turn. You should first think about the current condition and plan for your future actions, and then output your action in this turn. Your output must strictly follow this format:"Thought: your thoughts.\nAction: your next action".

The available actions are:
1. go to {recep}
2. task {obj} from {recep}
3. put {obj} in/on {recep}
4. open {recep}
5. close {recep}
6. toggle {obj} {recep}
7. clean {obj} with {recep}
8. heat {obj} with {recep}
9. cool {obj} with {recep}
where {obj} and {recep} correspond to objects and receptacles.
After your each turn, the environment will give you immediate feedback based on which you plan your next few steps. If the environment output "Nothing happened", that means the previous action is invalid and you should try more options.

Your response should use the following format:

Thought: <your thoughts>
Action: <your next action>

---

**Assistant**

OK

**User**

You are in the middle of a room. Looking quickly around you, you see a bathtubbasin 1, a cabinet 2, a cabinet 1, a countertop 1, a garbagecan 1, a handtowelholder 1, a sinkbasin 1, a toilet 1, a toiletpaperhanger 1, and a towelholder 1. Your task is to: put a toiletpaper in toiletpaperhanger.

**Assistant**

Thought: I need to place a toilet paper roll on the toilet paper hanger. First, I should check if there's already a roll on the hanger or if it's empty.
Action: go to toiletpaperhanger 1

**User**

Observation: On the toiletpaperhanger 1, you see nothing.

**Assistant**

Thought: The toilet paper hanger is empty, so I need to find a new roll of toilet paper to put on it.
Action: go to toilet 1

**User**

Observation: On the toilet 1, you see a soapbottle 1, and a toiletpaper 1.

**Assistant**

Thought: I've found a toilet paper roll on the toilet. I should pick it up so I can place it on the hanger.
Action: take toiletpaper 1 from toilet 1

**User**

Observation: You pick up the toiletpaper 1 from the toilet 1.

**Assistant**

Thought: Now that I have the toilet paper roll, I need to return to the toilet paper hanger to install it.
Action: go to toiletpaperhanger 1

**User**

Observation: On the toiletpaperhanger 1, you see nothing.

**Assistant**

Thought: I'm back at the toilet paper hanger with the roll in hand. It's time to put the toilet paper on the hanger.
Action: put toiletpaper 1 in/on toiletpaperhanger 1

## C.3. ScienceWorld Demonstration

To further illustrate the reasoning capabilities of LLMs across environments, we provide the initial prompt for a task in the ScienceWorld environment. For brevity, only the starting prompt is shown here; the subsequent interaction process is analogous to the ALFWorld demonstration, which is described in Sec. C.2, and is omitted.

---

**System**

You are a helpful agent that interacts with the virtual science school environment to solve the given task.

---

**User**

You are an agent for ScienceWorld. Every round I will give you an observation, and you have to respond with an action based on the observation to finish the given task. Here are the actions you may take:["action": "open/close OBJ", "description": "open/close a container", "action": "de/activate OBJ", "description": "activate/deactivate a device", "action": "connect OBJ to OBJ", "description": "connect electrical components", "action": "disconnect OBJ", "description": "disconnect electrical components", "action": "use OBJ [on OBJ]", "description": "use a device/item", "action": "look around", "description": "describe the current room", "action": "look at OBJ", "description": "describe an object in detail", "action": "look in OBJ", "description": "describe a container's contents", "action": "read OBJ", "description": "read a note or book", "action": "move OBJ to OBJ", "description": "move an object to a container", "action": "pick up OBJ", "description": "move an object to the inventory", "action": "put down OBJ", "description": "drop an inventory item", "action": "pour OBJ into OBJ", "description": "pour a liquid into a container", "action": "dunk OBJ into OBJ", "description": "dunk a container into a liquid", "action": "mix OBJ", "description": "chemically mix a container", "action": "go to LOC", "description": "move to a new location", "action": "eat OBJ", "description": "eat a food", "action": "flush OBJ", "description": "flush a toilet", "action": "focus on OBJ", "description": "signal intent on a task object", "action": "wait", "description": "take no action for 10 iterations", "action": "wait1", "description": "take no action for 1 iteration", "action": "task", "description": "describe current task", "action": "inventory", "description": "list your inventory"]
Your response should use the following format:
Thought:
your thoughts.

Action:
your next action

---

**Assistant**

OK. I'll follow your instructions and try my best to solve the task.

---

**User**

Your task is to boil water. For compounds without a boiling point, combusting the substance is also acceptable. First, focus on the substance. Then, take actions that will cause it to change its state of matter.
This room is called the hallway. In it, you see:
a picture
a substance called air
the agent
You also see:
A door to the green house (that is open)
A door to the living room (that is open)
A door to the art studio (that is open)
A door to the kitchen (that is open)
A door to the bedroom (that is open)
A door to the workshop (that is open)

*Table 10.* OPC-SFT hyperparameters for **ALFWorld** and **ScienceWorld**.

| Hyperparameter | ALFWorld | ScienceWorld |
|---|---|---|
| Clipping ratio $\epsilon$ | 0.5 | 0.5 |
| Learning rate | 1e-5 | 2e-6 |
| Rollout batch size | 256 | 256 |
| Train batch size | 32 | 32 |
| Maximum epochs | 3 | 3 |
| Number of episodes | 3 | 3 |
| Prompt maximum length | 4000 | 4000 |

*Table 11.* Reinforcement learning hyperparameters for **ALFWorld** and **ScienceWorld**.

| Hyperparameter | ALFWorld | ScienceWorld |
|---|---|---|
| Learning rate | 1e-6 | 1e-6 |
| KL loss coefficient | 0.01 | 0.01 |
| KL coefficient | 0.001 | 0.01 |
| KL loss type | Low Var KL | Low Var KL |
| Rollout temperature | 0.7 | 0.7 |
| Validation temperature | 0.7 | 0.7 |
| Maximum prompt length | 8192 | 8192 |
| Maximum response length | 256 | 128 |
| Clipping ratio low | 0.2 | 0.2 |
| Clipping ratio high | 0.2 | 0.2 |
| Rollout N | 8 | 8 |
| Max environment steps | 40 | 30 |
| PPO mini batch size | 16 | 32 |
| Max number of sequences | 512 | 1024 |
| Critic warm-up | 0 | 0 |

## C.4. Hyperparameters

In Tab. 10 and Tab. 11 we list the key hyperparameters required for both OPC-SFT and reinforcement learning.

## C.5. Out-of-Distribution Datasets

To provide a comprehensive evaluation of knowledge retention and generalization, we include several widely used benchmarks from different domains. Below we briefly describe each dataset:

**MBPP (Austin et al., 2021).** The Mostly Basic Python Problems (MBPP) dataset consists of 378 hand-written Python programming problems designed to evaluate models' ability to generate correct and efficient code. Each problem includes a description and a reference implementation, and performance is measured using functional correctness tests.

**HumanEval (Chen et al., 2021).** The HumanEval dataset provides 164 Python programming tasks accompanied by unit tests. It is commonly used to assess the code generation ability of large language models.

**MMLU (Hendrycks et al., 2021a).** The Massive Multitask Language Understanding (MMLU) benchmark evaluates broad general knowledge across 57 tasks covering mathematics, history, law, medicine, and other domains. It is designed to test both world knowledge and problem-solving ability.

**GPQA (Rein et al., 2023).** The Graduate-Level Google-Proof Q&A benchmark contains 198 challenging questions curated by subject matter experts, with a focus on requiring reasoning beyond simple retrieval.

**LiveCodeBench (Jain et al., 2025)** The LiveCodeBench benchmark evaluates live code generation and execution ability under dynamic environments. It provides 442 diverse programming challenges with runtime validation, extending beyond static unit-test benchmarks.

# D. LLM Fine-Tuning Related Work

LLMs have demonstrated a strong capacity for multi-step reasoning, a crucial component for solving complex problems (Zhao et al., 2023). This capability is rooted in their pre-training on extensive and diverse corpora (Qwen et al., 2025; Touvron et al., 2023). While high-quality pre-training data is critical for shaping these foundational abilities, it is often insufficient for specialized, challenging domains. Agentic tasks, for example, demand complex reasoning that is deeply integrated with planning and executing actions in an interactive environment (Wu et al., 2025a). Therefore, post-training is essential to adapt LLMs to these specific domains, significantly enhancing their ability to perform such intricate tasks (Wang et al., 2025; Team et al., 2025).

## D.1. Supervised Fine-Tuning

SFT is a foundational post-training stage that significantly enhances LLMs by aligning them with human instructions. By training on high-quality prompt-response pairs, SFT refines the model's ability to generate coherent and contextually appropriate outputs. However, this process introduces a critical trade-off: while extensive SFT improves instruction-following, it can also reduce the diversity of the model's generations (Li et al., 2025; Wang et al., 2024). Over-optimization on a fixed set of responses may lead to mode collapse, where the model consistently produces similar outputs, thereby limiting its exploratory capabilities. This loss of diversity is particularly detrimental for downstream reinforcement learning, where a broad search space is essential for discovering optimal policies (Zeng et al., 2025). Striking a balance between alignment and diversity is thus a key challenge, as excessive fine-tuning risks narrowing the model's generative flexibility. Indeed, recent studies suggest that SFT can substantially alter the LLM's latent space, limiting transferability (Huan et al., 2025). Consequently, some approaches bypass SFT entirely, using direct reinforcement learning to enhance exploration and improve reasoning (DeepSeek-AI et al., 2025; Zeng et al., 2025). There are also concurrent works seeking methods to enhance generalization for SFT (Zhu et al., 2025; Wu et al., 2025b). While our method shares the operational concept of integrating trust regions into SFT with concurrent work (Zhu et al., 2025), our motivation differs significantly. Our approach is driven by the insight that off-policy tokens, characterized by large gradient norms, are the primary drivers of catastrophic forgetting.

## D.2. Reinforcement Learning

Reinforcement learning provides a general framework for sequential decision problems (Sutton, 2018; Liu et al., 2025; Zhang et al., 2024b). Building on the framework of Reinforcement Learning from Human Feedback (RLHF), recent studies have extended RL to enhance the reasoning capabilities of LLMs (Trung et al., 2024; Kazemnejad et al., 2024; Gehring et al., 2024). Beyond its application in mathematical reasoning, RL provides a general mechanism for optimizing non-differentiable objectives, aligning models with human preferences, and encouraging effective exploration of solution spaces. By directly shaping model behavior through reward signals, RL complements supervised training and enables LLMs to achieve improved generalization and robustness. Nevertheless, applying standard algorithms such as PPO is resource-intensive, as it requires an additional critic network, substantially increasing computational cost and GPU memory usage. To alleviate this, ReMax (Li et al., 2024) employs the REINFORCE algorithm with greedy sampling as a reward baseline. Group Relative Policy Optimization (GRPO) (Shao et al., 2024) introduces a more memory-efficient variant of PPO that enhances reasoning performance. Reinforce++ (Hu, 2025) integrates techniques such as PPO clipping and reward normalization to improve stability and training efficiency. Furthermore, since policy entropy tends to diminish rapidly during training, reducing exploration, DAPO (Yu et al., 2025b) proposes the Clip-Higher strategy to counteract this effect.

## D.3. Continual Learning

Continual Learning (CL) aims to enable models to learn from a stream of tasks sequentially without suffering from catastrophic forgetting (Lopez-Paz & Ranzato, 2017; Kirkpatrick et al., 2017; De Lange et al., 2022). In the context of LLM fine-tuning, continual learning focuses on adapting models to a sequence of domains or instructions, with the goal of learning new capabilities without degrading performance on earlier tasks (Wu et al., 2024). In current mainstream approaches, replay-based methods alleviate forgetting by storing a small subset of data from previous tasks in a memory buffer and revisiting them during training to maintain historical knowledge (Scialom et al., 2022). Regularization-based approaches constrain parameter updates to prevent interference with knowledge acquired from previous tasks (Razdaibiedina et al., 2023). While continual learning typically targets the maintenance of performance across a sequence of distinct supervised tasks, our work primarily focuses on retaining general pre-trained capabilities, most notably reasoning and prior knowledge,

during the SFT cold-start phase. Consequently, replay is less suitable in our setting, since constructing a compact yet representative buffer for general abilities is non-trivial. We also compare against regularization-based baselines, including NEFTune (Jain et al., 2023), which improves robustness by injecting noise during fine-tuning.

