# OpenReview forum: "Clipping Low-Probability Tokens in SFT Yields a Generalizable Initialization for RL"
_ICML.cc/2026/Conference — ICML 2026 regular_

### Official Review · Reviewer_QZ6x · 2026-03-09

**Soundness:** 2
**Presentation:** 3
**Significance:** 2
**Originality:** 1
**Overall Recommendation:** 3
**Confidence:** 4

**Summary:**

The paper attributes SFT-induced catastrophic forgetting to low-probability target tokens producing large gradient norms.  It applies PPO-style ratio clipping to SFT and shows ~11.5% less OOD forgetting and ~7% better post-RL performance on ALFWorld/ScienceWorld with Qwen2.5-{1.5B,7B} and Llama3.2-3B.

**Compliance With Llm Reviewing Policy:**

Affirmed.

**Final Justification:**

I will maintain my score

**Key Questions For Authors:**

1. What is the contribution relative to PSFT
2. Why does OPC-SFT beat DFT, and what does this say about your diagnosis?

**Limitations:**

No limitation discussed. e.g. 1) Scale: capped at 7B;  2. domain diversity: primarily agentic  3. No theory

**Strengths And Weaknesses:**

**Strengths**
* Clean evaluation protocol (in-domain, OOD, post-RL) on a sensible agentic testbed.
*  Gains are largest when SFT data is highly off-policy

**Weaknesses**
1. Novelty. The loss is identical to PSFT, accepted at ICLR 2026, no longer "concurrent" as cited. The paper's only attempt at differentiation is a single sentence in Appendix C.1 stating that "our motivation differs significantly."  Motivation difference itself cannot the novelty of a whole paper
2. the motivation from PSFT seems natural to me, but the motivation form this paper leads more naturally to a different method (DFT). The diagnosis from this paper is: low-prob tokens produce large gradient norms and dominate early updates, causing forgetting. The natural fix this implies is to directly downweight low-prob tokens like DFT. This paper chooses to cut those large-ratio tokens, which might be a different set with low prob tokens
3. No theory.analysis

---

> ### Author Rebuttal · Authors · 2026-03-31
>
> Dear Reviewer QZ6x, thank you for your insightful comments. We hope that our rebuttal addresses your concerns.
>
> 1. Novelty.
>
> We agree that novelty should not be claimed from the clipping strategy itself, and we will revise the paper to make this point clearer and to explain the difference from PSFT more explicitly. The contribution of our work lies in the findings on why forgetting happens, why clipping works, and when it is most useful. Our work starts from the finding that low-probability target tokens induce large gradient norms, which then cause large probability shifts and lead to forgetting of pre-trained knowledge. Based on this finding, we aim to postpone overly aggressive updates on these low-probability tokens in order to preserve prior knowledge and improve generalization. A clipping strategy naturally provides a proximal mechanism for limiting the magnitude of such aggressive updates. Since the clipping strategy itself is not newly proposed by either paper, both PSFT and our work adopt a similar tool and analyze how it can improve learning. The key difference is therefore not the clipping operator itself, but the findings on how and when it should be used.
>
> PSFT provides an intuitive motivation and evaluates the method on math tasks, where, as shown in both papers, the gain is not significant. In contrast, our work provides a more specific set of findings: we connect low-probability tokens to large gradient norms, and further connect these large gradients to forgetting and reduced generalization retention. Based on these findings, we show that clipping is particularly effective in the agentic setting, where the initial gradient norms are large and the data distribution is less familiar to the base model. This analytical distinction also leads to a different intended regime. PSFT discusses an optional warm-up stage before clipping, whereas our analysis suggests that the most harmful forgetting may already happen during that stage. This is because the warm-up phase may already damage prior knowledge, so by the time clipping is applied, forgetting may have already occurred.
>
> 2. Low-prob Tokens and Large Ratio.
>
> Although importance reweighting like DFT is a straightforward method to restrain the learning process, our goal is not to ban the learning of low-probability tokens, but to learn them in a prioritized, progressive manner.
> As shown in our Figure 1(c), some crucial task-specific tokens have initial probabilities as low as $10^{-4}$. If we aggressively suppress them through static probability reweighting, their consistently small gradient norms will prevent the model from learning them effectively, resulting in suboptimal task performance.
> Instead, our clipping strategy acts as a dynamic speed limit. Early in training, the clipped tokens largely overlap with the low-probability ones. By capping the ratio $\frac{\pi_{\theta}}{\pi_{\theta_{\text{old}}}}$, clipping prevents catastrophic forgetting. As learning proceeds and the model gradually acquires these tokens, their probabilities increase, and the ratio shrinks, as shown in Figure 11. This progressive process allows the model to learn necessary new knowledge while retaining generalization.
>
> 3. Theory.
>
> We show that making a target token highly likely when its original probability is very low requires a large increase in the target-token margin $\Theta(\log(1/\delta))$, where $\delta:=\pi_{\theta_{\mathrm{old}}}(y^{\ast}\mid x)\ll 1$. Under a local smoothness condition, this also implies a large movement in parameter space. By contrast, OPC-SFT clips the policy ratio within each reference interval, which bounds the corresponding margin increase per interval to $O(\log(1+\epsilon))$, where $\epsilon$ is the clipping threshold. Therefore, OPC-SFT turns a potentially abrupt large correction into smaller controlled updates, reducing representational drift and better preserving the model’s pre-trained generalization ability. The complete proof is shown in [link](https://anonymous.4open.science/r/figure-4A1D/proof.png).
>
>
> 4. Larger Base Model.
>
> Qwen3-32B training result is shown in [link](https://anonymous.4open.science/r/figure-4A1D/32b.png).
>
>
> 5. More Domains.
>
> We present general-domain instruction following results on AlpacaEval 2.0 for models trained on an OpenOrca subset. These results show that OPC-SFT remains effective in this standard fine-tuning setting.
>
>
> | Backbone | Method | In-Distribution (AlpacaEval 2.0) After RL | | OOD (Code) Before RL | |
> | :--- | :--- | :---: | :---: | :---: | :---: |
> | | | LC Win Rate | Win Rate  | HumanEval | MBPP |
> | **Qwen2.5-7B** | SFT | 18.4 | 16.2 | 58.0 | 31.7 |
> | | NEFT | 19.5 | 17.1 | 61.2 | 33.1 |
> | | DFT | 20.2 | 18.5 | 59.5 | 32.3 |
> | | OPC-SFT | **21.3** | **19.8** | **63.5** | **34.1** |
> | **Llama3.1-8B** | SFT | 21.6 | 18.5 | 48.0 | 35.7 |
> | | NEFT | 21.0 | 19.8 | 48.8 | 36.2 |
> | | DFT | 20.1 | 19.2 | 48.4 | 36.8 |
> | | OPC-SFT | **22.8** | **20.3** | **49.1** | **37.3** |

---

> > ### Author Rebuttal · Reviewer_QZ6x · 2026-04-03
> >
> > I thank the authors for the rebuttal and the additional results. This partially addresses my concerns but not fully.
> >
> > The authors concede the clipping strategy itself is not novel and position the analysis (low-π tokens → large gradients → forgetting) as the contribution. However, this analysis only tells us *which* tokens are problematic (guide to DFT); it does not tell us *how* to handle them. The step from "low-π tokens" to "clip large-ratio tokens against a moving π_old" is exactly the trust-region argument of PSFT, and is the part that actually makes the method work.
> >
> > So the claimed contribution is neither necessary nor sufficient (the analysis alone leads to DFT, not to ratio clipping) for the method.

---

> > > ### Author Response · Authors · 2026-04-07
> > >
> > > We thank the reviewer for the time and effort in evaluating our work. We fully understand this concern and aim to clarify how our contribution is distinguished from existing perspectives. To this end, we respond in two parts below. We sincerely hope this clarification better conveys the main value of our work, and we would be very grateful for your reconsideration.
> > >
> > > 1. Relation to Token-Probability Reweighting.
> > >
> > > We would like to clarify why our low-probability diagnosis can indeed motivate a clipping-based intervention. For a target token $y^{\*} $, let$ p:=\pi_{\theta}(y^{\*} \mid x ), q := \pi_{\theta_{{\mathrm{old}}}}(y^{\*} \mid x)$
> > > and define the probability change $\Delta := p - q.$
> > > Then the policy ratio used in OPC-SFT can be written exactly as:
> > > $$
> > > r_\theta = \frac{p}{q} = 1 + \frac{\Delta}{q}.
> > > $$
> > > So the clipping threshold $r_\theta \le 1+\epsilon$ is equivalent to
> > > $
> > > \Delta \le \epsilon q.
> > > $
> > > This gives a precise interpretation of OPC-SFT: within each old-policy refresh window, each token is assigned an absolute probability-increase budget bounded by $\Delta \le \epsilon q$. As a result, tokens with smaller old-policy probability $q$ are allowed a smaller absolute change in a single update interval, while tokens with larger $q$ can move more. This interpretation is also consistent with our token-level statistics shown in [[link]](https://anonymous.4open.science/r/figure-4A1D/token_clip_distribution.png), where clipped tokens substantially overlap with low-probability tokens. Thus, clipping mainly acts on the same token subset highlighted by our diagnosis, namely those most associated with overly rapid early drift. In addition, low-$q$ tokens are not excluded from learning, but are prevented from making abrupt early jumps, while still allowing progressive adaptation across refresh windows.
> > >
> > > This also explains why OPC-SFT should be viewed as a different form of importance reweighting. DFT applies an absolute probability weighting, effectively scaling each token only by its current probability $p$. By contrast, OPC-SFT applies an old-policy-normalized and truncated weighting, with an effective factor:
> > >
> > > $$
> > > w_{\mathrm{OPC}}(p,q)=\frac{p}{q} \cdot \mathbb{1}[p \le (1+\epsilon)q].
> > > $$
> > >
> > > Therefore, DFT persistently downweights tokens whenever their current probability is low, whereas OPC-SFT intervenes only when a token moves too rapidly relative to the old policy. This distinction is important for our setting. Our empirical finding is that forgetting is most severe at the initial stage of fine-tuning, when a subset of low-probability targets can induce abrupt updates and rapid drift. From this perspective, clipping is more consistent with our findings than static probability reweighting, as OPC-SFT constrains this kind of overly rapid early drift through the condition $\Delta \le \epsilon q$.
> > >
> > > 2. Relation to PSFT and Trust-Region Motivation.
> > >
> > > PSFT provides a useful trust-region view of clipped SFT by borrowing PPO/TRPO-style intuition. We respectfully disagree that the PSFT trust-region argument is exactly what makes the method work. In our view, the trust-region argument mainly describes what the clipping operator does: it bounds the change relative to an old policy. However, this does not explain why such control is especially needed in SFT, nor why the ratio is the right quantity for deciding which tokens should be clipped. In PPO-style RL, the ratio is used because updates reuse samples collected under an older policy. In SFT, by contrast, supervision comes from a fixed expert dataset rather than rollouts of $\pi_{\mathrm{old}}$, so this motivation does not transfer in the same direct way. Our work precisely fills this gap. We show that catastrophic forgetting in SFT is concentrated at the earliest stage of fine-tuning and is disproportionately driven by low-probability target tokens, which induce unusually large gradients and overly rapid token-level drift.
> > >
> > > Empirically, both PSFT and our work show that clipped SFT is at least competitive on math, but the gains there are not especially large. Our work goes one step further by analyzing the gradient distribution at the beginning of SFT and identifying a setting where clipped SFT is more effective. In this setting, low-probability target tokens induce unusually large gradient norms, which drive rapid early drift and forgetting. This explains why clipping is especially effective here, and why the gains are much more visible on ALFWorld and ScienceWorld than on math. To our knowledge, our work is the first to identify and explain this phenomenon, and to empirically demonstrate clear gains in off-policy agentic tasks.
> > >
> > > Regularized SFT is increasingly regarded as important, since the gradient norms at the start of SFT are often very large and can disturb the model. We believe it is also valuable to provide a deeper analysis of when and how clipping can be used as a reasonable tool, and we hope this clarification will be helpful for your final assessment.

---

### Official Review · Reviewer_JKuv · 2026-03-12

**Soundness:** 3
**Presentation:** 3
**Significance:** 2
**Originality:** 2
**Overall Recommendation:** 4
**Confidence:** 3

**Summary:**

This paper investigates knowledge forgetting during the SFT phase and proposes a method to mitigate it. The authors first observe that SFT produces larger probability shifts than RL on OOD tasks, and forgetting is primarily concentrated in the early training epochs. Further analysis reveals that tokens the policy model assigns low probability produce disproportionately large gradients, driving overly aggressive parameter updates in the early stage and leading to rapid degradation of OOD knowledge. Based on this finding, the authors draw inspiration from the clipping mechanism in PPO and propose OPC-SFT: compute the probability ratio between the current and old policies for each token, and set the gradient to zero when the ratio exceeds a threshold, thereby limiting the magnitude of single-step updates. Experiments on Qwen2.5-7B, Qwen2.5-1.5B, and Llama3.2-3B show that OPC-SFT matches or slightly outperforms baselines on in-domain tasks, reduces average OOD performance degradation, and improves final RL performance. OPC-SFT produces smaller latent-space shifts and smoother token probability trajectories. Ablation studies verify the robustness of the clipping ratio and old policy update frequency.

**Compliance With Llm Reviewing Policy:**

Affirmed.

**Final Justification:**

The paper offers a clear analytical chain and thorough experiments. The rebuttal addressed several concerns constructively: the LR decay baseline confirms that OPC-SFT's benefit is not reducible to simple learning-rate scheduling; the clipped-token analysis shows that clipping targets task-relevant tokens; and the Qwen3-32B result extends the validation scope. These additions yield useful insights that complement existing work, like DFT and PSFT. However, methodological overlap with concurrent work persists, and task diversity remains limited. Balancing the solid experimental work against the limited originality, I slightly raise my score.

**Key Questions For Authors:**

1. The paper analysis identifies low-probability tokens as the root cause of forgetting, but OPC-SFT clips tokens with large ratios rather than tokens with low probabilities. Can you provide experimental evidence demonstrating that these two token categories overlap substantially? If they do not fully overlap, does this imply that clipping may inadvertently penalize some tokens that should be learned?

2. The proportion of clipped tokens approaches zero in the later stages of training. Does this mean that OPC-SFT's primary effect is limited to the first few training steps? If so, could a simple baseline of "reducing the learning rate for the first few steps, followed by normal SFT" achieve a similar effect?

3. How does OPC-SFT perform on larger-scale or more recent, stronger models? Currently, validation is limited to Qwen2.5 and Llama3.2 on agentic tasks. I believe that supplementing experiments with larger model sizes or reasoning models would better support the method's generality.

**Limitations:**

yes

**Strengths And Weaknesses:**

Strengths:
1. From the comparison of probability shifts between SFT and RL, to the analysis linking gradient norms and token probabilities in the early training stage, to the gradient norm statistics across probability bins, the paper presents a clear and complete causal chain throughout.
2. The experimental design covers three dimensions: in-distribution generalization, OOD generalization, and final post-RL performance, supplemented by PCA analysis, token probability tracking, and ablation of the clipping component, providing multi-faceted verification of the method's mechanisms.
3. OPC-SFT introduces only two additional components, the clipping ratio and old policy update frequency, making it simple to implement and seamlessly compatible with existing training pipelines. Meanwhile, compared to standard SFT and DFT, it demonstrates clear improvements after RL, offering good practical engineering value.

Weaknesses:
1. The core method of this paper, applying a clipping mechanism to SFT, has significant overlap with some existing works. First, the observation that low-probability tokens produce large gradients leading to forgetting is not entirely novel. DFT has already proposed using token probability reweighting of the loss to address this issue. Furthermore, PSFT [1], which is mentioned in the paper but not discussed in detail, proposes similar formulations and systematically analyzes entropy collapse and warm-up strategies. iw-SFT [2] has also derived importance-weighted SFT from the perspective of the RL lower bound and leverages clipping.

2. The analysis section of the paper focuses on "low-probability tokens producing large gradient norms," but the proposed method OPC-SFT clips based on the policy ratio rather than directly targeting low-probability tokens. A token whose probability changes from 0.01 to 0.015 has a ratio of 1.5, while a token whose probability changes from 0.5 to 0.7 has a ratio of 1.4. Besides, the paper starts from PPO's clipping and sets the advantage to a constant of 1. This simplification lacks sufficient justification and assumes all tokens in the expert data are equally important.

3. The analysis of key hyperparameters is insufficiently thorough. The old policy update frequency and clipping threshold are core hyperparameters of this method, but the ablation experiments test only 4–5 values and are conducted on a single model and a single task. Given the relatively low cost of SFT compared to RL, I think a few additional experimental configurations should be added.

4. There is no detailed analysis of the clipped tokens. The paper lacks an analysis of the "semantic types of clipped tokens". Are the clipped tokens noise tokens or critical task-specific tokens? Furthermore, the appendix shows that the clipping ratio drops rapidly to near zero during training, indicating that clipping barely takes effect in later stages and that the method appears to degenerate into something close to standard SFT.

[1] Proximal Supervised Fine-Tuning, 202508

[2] Supervised Fine-Tuning on Curated Data is Reinforcement Learning, 202507

---

> ### Author Rebuttal · Authors · 2026-03-31
>
> Dear Reviewer JKuv, thank you for your insightful comments. We hope our rebuttal addresses your concerns.
>
> 1. Novelty.
>
> We agree that the clipping operator itself is not the novelty, and we will make this clearer and to discuss the difference from PSFT explicitly. Since clipping is not a novel technique and is adopted just as a tool, the key difference lies in the analysis: our work identifies that low-probability target tokens induce large gradient norms, which accelerate forgetting of prior knowledge and degrade generalization. This analysis also clarifies that clipping is most useful when the SFT data is off-policy and the initial gradients are large, as in agentic settings. By contrast, PSFT mainly studies math tasks, where the data is closer to the LLM’s pre-training distribution and the gains from clipping are less pronounced. This difference in analysis also implies a different intended regime: while PSFT includes an optional warm-up stage before clipping, our results suggest that harmful forgetting may already occur during that stage.
>
> Compared with importance reweighting methods such as DFT, our goal is to learn the low-prob tokens in a more progressive manner. As shown in Figure 1(c), some important task-specific tokens start with probabilities as low as $10^{-4}$. If they are consistently downweighted, their gradients may remain too small for effective learning. In contrast, our clipping strategy acts as a dynamic speed limit: it mainly constrains these tokens early in training, when their updates are most aggressive, and relaxes automatically as they are gradually learned (Figure 11). This yields a more friendly learning process that balances adaptation and generalization.
>
> 2. Detailed Analysis of the Clipped Tokens.
>
> We analyzed the semantic meaning of the clipped tokens in [link](https://anonymous.4open.science/r/figure-4A1D/realistic_clip_wordcloud_purple.png). Many clipped tokens are not noise and clearly carry task-specific meaning. Our point is therefore not that these tokens should be excluded from learning, but that they should not be updated too aggressively at the beginning. The model should first learn relatively familiar tokens and then gradually adapt to unfamiliar ones.
>
> The declining proportion of clipped tokens is also expected. It indicates that clipping mainly acts in the early stage, when the model is still far from the target distribution. As training proceeds and the model becomes better aligned with the task-critical data, fewer tokens require clipping.
>
> 3. Token Probability and Policy Ratio.
>
> We observe a significant overlap between the clipped tokens and the low-prob tokens in [link](https://anonymous.4open.science/r/figure-4A1D/token_clip_distribution.png). A direct idea is to operate on their probabilities, such as using importance reweighting like DFT. However, our comparisons reveal that the clipping ratio is more robust. Absolute probabilities often lack clear numerical meaning for guiding method design, and directly manipulating probabilities is fragile for critical task-specific tokens with extremely low initial probabilities, e.g., $10^{-4}$.
>
> Our framework is grounded in Maximum Likelihood Estimation (MLE), and we naturally follow the SFT paradigm by assigning a uniform weight of 1 to all tokens, without prioritizing specific ones. The policy ratio is introduced strictly as a clipping mechanism to bound large updates, independent of the PPO setting.
>
> 4. LR Baseline.
>
> We test an additional baseline that reduces the learning rate to $1e-6$ only in the early stage of SFT. The results show that OPC-SFT still clearly outperforms this baseline. A lower learning rate uniformly shrinks the updates of all tokens, regardless of whether they are risky or already well aligned, which only slows learning globally. In contrast, OPC-SFT acts selectively: it constrains those tokens whose updates are aggressive relative to the current policy, while leaving other tokens much less affected. Therefore, the declining clipping ratio reflects a progressive and selective learning process, rather than a degeneration into standard SFT.
> ||Method|Before RL (ALFWorld)||After RL (ALFWorld)||
> |:---|:---|:---:|:---:|:---:|:---:|
> |||seen|unseen|seen|unseen|
> |Llama3.2-3B-Instruct|LR_decay|72.86|73.13|86.43|87.31|
> ||OPC-SFT|**76.43**|**77.61**|**94.29**|**92.54**|
> |Qwen2.5-1.5B-Instruct|LR_decay|**73.57**|71.64|87.86|91.04|
> ||OPC-SFT |72.86|**72.39**|**90.00**|**94.03**|
> |Qwen2.5-7B-Instruct |LR_decay|79.29|75.37|90.71|86.57|
> ||OPC-SFT|**82.86**|**78.36**|**92.14**|**91.04**|
>
> 5. Ablation Study.
>
> We conduct ablation experiments on more values and three models. Results are shown in [clip_ablation](https://anonymous.4open.science/r/figure-4A1D/clip_ratio_ablation.png) and [freq_ablation](https://anonymous.4open.science/r/figure-4A1D/update_frequency_ablation.png).
>
> 6. Larger and More Recent Model.
>
> Qwen3-32B training result is shown in [link](https://anonymous.4open.science/r/figure-4A1D/32b.png).

---

> > ### Author Rebuttal · Reviewer_JKuv · 2026-04-04
> >
> > Some of my concerns are resolved. Please include the findings reported in the rebuttal in the final version of the paper. Although the claims regarding novelty still fail to articulate the fundamental differences from other existing works, I will raise my score.

---

> > > ### Author Response · Authors · 2026-04-07
> > >
> > > We sincerely thank the reviewer for the thoughtful follow-up, for recognizing the value of our additional analyses and results, and especially for raising the score. We will incorporate these rebuttal findings into the final version as requested. We also understand that the remaining concern is how our contribution should be more clearly distinguished from related lines of work.
> > >
> > > The contribution of our work lies in identifying why forgetting happens in SFT, why clipping works in this setting, and when it becomes especially useful. Our analysis starts from the finding that low-probability target tokens induce unusually large gradient norms, which in turn cause rapid probability shifts and early forgetting of pre-trained knowledge. Based on this finding, our goal is not to prevent the learning of such tokens, but to postpone overly aggressive updates on them so that the model can preserve prior knowledge while adapting progressively. In this sense, clipping is a natural mechanism for limiting abrupt early movement caused by those low-probability tokens, and the key difference from prior work lies not in the clipping operator itself, but in the insight into how and when it should be used.
> > >
> > > From this perspective, we view our work not as overlapping with PSFT, but as a fundamental analysis that complements it. PSFT provides a useful proximal or trust-region view of clipped SFT, but in our view, this mainly describes what the operator does, rather than fully explaining why such control is especially needed in SFT. In PPO-style RL, the ratio is motivated by the reuse of samples collected under an older policy, whereas in SFT, supervision comes from a fixed expert dataset, so the motivation does not carry over in the same direct way. Our work fills this gap by providing an SFT-specific mechanism: catastrophic forgetting is concentrated at the earliest stage of fine-tuning and is disproportionately driven by low-probability target tokens with large gradients. This also leads to a clearer empirical picture. Both PSFT and our work show that clipped SFT is at least competitive on math, but the gains there are not especially large. Our work goes one step further by analyzing the initial gradient distribution and identifying that clipped SFT is much more effective in highly off-policy agentic cold-start settings. In this setting, the data distribution is less familiar to the base model, the early gradients are larger, and clipping yields much clearer gains. Our method also differs from static importance reweighting methods such as DFT in that DFT persistently downweights low-probability tokens throughout training, whereas our method promotes more progressive learning by preventing abrupt early updates while still allowing those tokens to be learned over time.

---

### Official Review · Reviewer_pe7o · 2026-03-12

**Soundness:** 3
**Presentation:** 3
**Significance:** 3
**Originality:** 3
**Overall Recommendation:** 4
**Confidence:** 3

**Summary:**

This paper identifies that low-probability "off-policy" tokens in SFT data generate disproportionately large gradient magnitudes, driving catastrophic forgetting of pre-trained knowledge. The authors propose OPC-SFT, which applies PPO-style token-level clipping to bound per-token updates during SFT. Experiments on agentic benchmarks across three model families show OPC-SFT reduces OOD forgetting by 11.54% and improves downstream RL performance by 7.09% over standard SFT.

**Compliance With Llm Reviewing Policy:**

Affirmed.

**Final Justification:**

I appreciate the authors’ response. The new rebuttals have alleviated my concerns about the novelty to some extent. Therefore, I will raise my score to 4.

**Key Questions For Authors:**

See Weaknesses.

**Limitations:**

Yes

**Strengths And Weaknesses:**

**Strengths**
1. The motivation is clear and practically relevant.

2. The authors conducted extensive experiments to evaluate the proposed method.


**Weaknesses**
1. Limited novelty. PSFT  proposes essentially the same mechanism. The paper's claimed differentiation is motivational rather than technical, and PSFT is never included as a direct experimental baseline.

2. The strong results are almost entirely driven by highly off-policy agentic tasks; the authors themselves show gains are modest on math reasoning. It remains unclear whether OPC-SFT provides meaningful benefits in more typical fine-tuning scenarios.

3. The paper dismisses KL-penalty methods as too costly without empirical comparison, leaving it unclear whether any distributional constraint on SFT would achieve similar results, or whether the clipping mechanism specifically is necessary.

4. Tables 4 & 5 only compare OPC-SFT against standard SFT on math tasks, omitting DFT, NEFT, and PSFT baselines that appear in the agentic evaluations. This makes it impossible to assess whether OPC-SFT's modest math gains are genuinely competitive among other methods.

---

> ### Author Rebuttal · Authors · 2026-03-31
>
> Dear Reviewer pe7o, thank you for your insightful comments. We hope that our rebuttal will address your concerns.
>
> 1. Novelty
>
> We agree that novelty should not be claimed from the clipping strategy itself, and we will revise the paper to make this point clearer and to explain the difference from PSFT more explicitly. The contribution of our work lies in our findings on why forgetting occurs, why clipping helps, and when it is most effective. Our work starts from the observation that low-probability target tokens induce large gradient norms, which in turn cause sharp probability shifts and lead to forgetting of pre-trained knowledge. Based on this finding, the goal is to postpone overly aggressive updates on these low-probability tokens so as to preserve prior knowledge and improve generalization. A clipping strategy naturally provides a proximal mechanism for limiting the magnitude of such updates.
>
> Since the clipping strategy itself is not newly proposed in either paper, both PSFT and our work adopt it as a tool and study how it can improve learning. The key difference, therefore, does not lie in the clipping operator itself, but in the findings about how and when it should be applied. PSFT mainly provides an intuitive motivation and evaluates the method on math tasks, where, as shown in both papers, the gains are relatively modest. In contrast, our work provides a more specific chain of evidence: we connect low-probability tokens to large gradient norms, and further connect these large gradients to forgetting and weaker retention of generalization. Based on these findings, we show that clipping is particularly effective in the agentic setting, where the initial gradient norms are larger, and the data distribution is less familiar to the base model. This analytical distinction also leads to a different intended regime. PSFT discusses clipping after an optional warm-up stage, whereas our analysis suggests that the most harmful forgetting may already occur during that stage. The reason is that the warm-up phase itself can already damage prior knowledge, so forgetting may have already taken place before clipping is activated. We additionally include the optional PSFT warm-up setting as a baseline.
>
> |Model|Method|Before RL Seen|Before RL Unseen|After RL Seen|After RL Unseen|
> |:---|:---|:---:|:---:|:---:|:---:|
> |**Llama3.2-3B-Instruct**|PSFT$_\textit{warm up}$|**77.85**|71.64|86.43|80.60|
> ||OPC-SFT|76.43|**77.61**|**94.29**|**92.54**|
> |**Qwen2.5-1.5B-Instruct**|PSFT$_\textit{warm up}$|**74.28**|71.64|84.29|85.82|
> ||OPC-SFT|72.86|**72.39**|**90.00**|**94.03**|
> |**Qwen2.5-7B-Instruct**|PSFT$_\textit{warm up}$|82.14|77.61|89.29|88.06|
> ||OPC-SFT|**82.86**|**78.36**|**92.14**|**91.04**|
>
> |Model|Method|MBPP|MMLU|HumanEval|GPQA|LiveCodeBench|MATH500|
> |:---|:---|:---:|:---:|:---:|:---:|:---:|:---:|
> |**Llama3.2-3B-Instruct**|PSFT$_\textit{warm up}$|58.47|59.50|**48.70**|17.17|40.50|36.80|
> ||OPC-SFT|**58.71**|**59.97**|48.13|**18.18**|**42.60**|**37.60**|
> |**Qwen2.5-1.5B-Instruct**|PSFT$_\textit{warm up}$|45.50|58.21|44.59|32.83|**33.14**|23.80|
> ||OPC-SFT|**46.56**|**58.85**|**44.96**|**33.84**|32.81|**24.20**|
> |**Qwen2.5-7B-Instruct**|PSFT$_\textit{warm up}$|78.04|70.15|74.69|33.84|66.52|71.80|
> ||OPC-SFT|**78.84**|**70.60**|**75.07**|**34.34**|**67.69**|**72.40**|
>
> 2. More Typical Fine-Tuning Scenarios.
>
> For typical fine-tuning scenarios such as instruction following for a base model, we add an experiment on general-domain instruction tuning trained with a 60k subset from the OpenOrca dataset and evaluate with AlpacaEval 2.0, as shown below. The results suggest that OPC-SFT remains helpful in this more standard fine-tuning setting.
> |Backbone|Method|In-Distribution (AlpacaEval 2.0) After RL||OOD (Code) Before RL||
> |:---|:---|:---:|:---:|:---:|:---:|
> |||**LC Win Rate**|**Win Rate**|**HumanEval**|**MBPP**|
> |**Qwen2.5-7B**|SFT|18.4|16.2|58.0|31.7|
> ||NEFT|19.5|17.1|61.2|33.1|
> ||DFT|20.2|18.5|59.5|32.3|
> ||OPC-SFT|**21.3**|**19.8**|**63.5**|**34.1**|
> |**Llama3.1-8B**|SFT|21.6|18.5|48.0|35.7|
> ||NEFT|21.0|19.8|48.8|36.2|
> ||DFT|20.1|19.2|48.4|36.8|
> ||OPC-SFT|**22.8**|**20.3**|**49.1**|**37.3**|
>
> 3. KL Penalty.
>
> We have added a KL-penalty baseline to our experiments. Since the exact KL is not tractable in this setting, we approximate it using the empirical expectation over the dataset during training. Although this regularizes policy updates, our results show that it does not effectively improve out-of-distribution generalization and also hurts learning efficiency. The result is shown in [link](https://anonymous.4open.science/r/figure-4A1D/KL_penalty.png).
>
> 4. Math Reasoning Result.
>
> We have now added the missing baselines. The results show that OPC-SFT remains competitive, although the gain is smaller than in the agentic setting. The result is shown in [link](https://anonymous.4open.science/r/figure-4A1D/math.png).

---

> > ### Author Rebuttal · Reviewer_pe7o · 2026-04-04
> >
> > Thank you for the detailed rebuttal. While some of my concerns have been addressed, I still have some doubts regarding the novelty of the paper. Therefore, I will keep my original score.

---

> > > ### Author Response · Authors · 2026-04-07
> > >
> > > We sincerely thank the reviewer for the valuable time and thoughtful suggestions on our work. We understand that the remaining concern is whether our paper provides sufficient insight beyond existing clipped SFT methods. We will revise the manuscript to make this positioning much more explicit, and we would be very grateful for your reconsideration.
> > >
> > > As also reflected in the final justifications of reviewers JKuv and ppip, this direction itself is important: the community has been looking for effective ways to regularize SFT, since the large gradients at the beginning of fine-tuning can significantly disturb the model and adversely affect downstream pipelines. That is to say, a complementary analysis is also valuable if it helps clarify how and when such a strategy should be used.
> > > Our claim is not that the clipping strategy itself is novel. Rather, our main contribution is a deeper analysis of why forgetting happens in SFT, why clipping works in this setting, and when it is especially useful. Concretely, our analysis starts from the finding that low-probability target tokens induce unusually large gradient norms, which in turn cause rapid probability shifts and early forgetting of pre-trained knowledge. We further verify that the tokens clipped by OPC-SFT substantially overlap with these low-probability targets, as shown in [[link]](https://anonymous.4open.science/r/figure-4A1D/token_clip_distribution.png). Figure 11 in our main text also shows that OPC-SFT acts selectively and progressively during training: it mainly constrains those tokens whose updates are overly aggressive relative to the current policy, while leaving other tokens much less affected.
> > >
> > > From this perspective, we do not view our work as overlapping with PSFT, but as a complementary analysis that explains when and why the clipping strategy works. PSFT provides a useful trust-region view of clipped SFT by borrowing PPO/TRPO-style intuition. However, in our view, this mainly describes what the clipping operator does; it does not by itself explain why such control is especially needed in SFT, or why the ratio is the right quantity for deciding which tokens should be clipped. In PPO-style RL, the ratio is motivated by reusing samples collected under an older policy, whereas in SFT, supervision comes from a fixed expert dataset, so this motivation does not carry over in the same direct way. Our work fills this gap by showing that catastrophic forgetting is concentrated at the earliest stage of fine-tuning and is disproportionately driven by low-probability target tokens with large gradients.
> > >
> > > The difference in motivation also leads to a different view of the operation itself, especially regarding warm-up. PSFT discusses an optional warm-up stage before clipping. However, Figures 1(b) and 1(c) in our main text show that the main damage to generalization already happens at the very beginning of fine-tuning, when the gradient norms are largest, and the target-token probabilities change most rapidly. Therefore, our analysis suggests that the warm-up may have already passed through the most harmful stage before clipping is applied. In this sense, our analysis not only complements PSFT but also helps explain why PSFT with warm-up can significantly degrade the original results.
> > >
> > > This deeper analysis also leads to a clearer empirical picture. Both PSFT and our work show that clipped SFT is at least competitive on math, but the gains there are not especially large. Our work goes one step further by analyzing the initial gradient distribution and identifying off-policy agentic cold-start as a setting where clipped SFT is much more effective. In this setting, the data distribution is less familiar to the base model, the early gradients are larger, and clipping yields much clearer gains on ALFWorld and ScienceWorld than on math. To our knowledge, our work is the first to identify and explain this phenomenon, and to empirically demonstrate clear gains in off-policy agentic tasks.

---

### Official Review · Reviewer_ppip · 2026-03-24

**Soundness:** 3
**Presentation:** 2
**Significance:** 3
**Originality:** 3
**Overall Recommendation:** 5
**Confidence:** 4

**Summary:**

This paper deals with the forgetting problem in SFT-then-RL pipelines. When you fine-tune on a small set of expert demonstrations, the model memorizes those traces and wrecks its general capabilities, giving RL a weaker starting point.

The core observation is straightforward. The log-likelihood gradient divides by the model's predicted probability for each target token. When the base model assigns very low probability to certain tokens in the expert data -- things like unfamiliar action syntax it never saw in pretraining -- the gradient blows up. Tokens in the lowest probability bin have average gradient norms around 32 versus about 2 for high-confidence tokens. So a small subset of tokens drives massively outsized updates that drag the model's representations away from the base model early in training.

The fix borrows PPO's clipping mechanism for supervised fine-tuning. You maintain a reference old policy, compute a probability ratio for each token, and if the current policy has moved too far from the reference on that token, the gradient gets attenuated. The old policy refreshes periodically so the trust region slides forward. The model can still learn from off-policy tokens, just not in huge jumps.

Results on ALFWorld and ScienceWorld across three backbones show about 11.5% less OOD forgetting and 7% better final RL performance after GRPO. The PCA analysis confirms the model's representations stay much closer to the base model. Gains are most pronounced on agentic tasks where the expert data is most foreign to the model -- math reasoning data is already fairly on-policy so there's less to fix. Ablations show it's specifically the clipping, not just the implicit ratio-based reweighting, that preserves prior knowledge.

**Compliance With Llm Reviewing Policy:**

Affirmed.

**Final Justification:**

Fundamentally, a way of doing regularized SFT is something that the community has been looking for for a while. It is well known that the grad norms at the start of SFT is massive, and they tend to move the model whenever a new task is started to be trained, and this can have pretty adverse impact on downstream pipelines. A reasonably principled way of doing selectively regularized SFT to mitigate the initial massive grad norms is great to see, other than off the shelf methods that affect all tokens equally.

The new rebuttals remove certain concerns about baselines and stronger benchmarks. This increases my score to a 5 due to the usefulness of this method.

**Key Questions For Authors:**

The paper dismisses KL-penalty baselines as costly without actually comparing against them. But KL regularization targets exactly the same problem -- constraining distributional shift from a reference policy -- and is standard practice. Can you provide either empirical comparison or a more concrete argument for why clipping is preferable beyond implementation convenience during SFT?

For the RL improvements, is the improvement purely from a better initialization that carries through, or is there evidence that the OPC-SFT initialization actually enables different exploration behavior or learning dynamics during RL that a standard SFT initialization couldn't achieve?

**Limitations:**

Yes.

**Strengths And Weaknesses:**

Soundness: The connection between low-probability tokens, large gradients, and early forgetting is well-motivated and empirically solid. The gradient norm analysis across probability bins is a clean experiment that makes the mechanism intuitive -- roughly, you can see exactly where the destructive updates come from. The fact that they track this across training steps and show most forgetting happens in the first epoch adds to more credibility to the overall story. The experimental design covers the right  evidence -- three model backbones, two agentic benchmarks, both in-distribution and OOD evaluation, and the full SFT-then-RL pipeline. The PCA latent space analysis adds a decent insight source beyond just reporting task metrics. overall a 3.

Presentation: The preliminaries section is longer than it needs to be, it is a bit verbose. The RL framing with state spaces and action spaces takes up space but does not really contribute much to understanding the actual method, which is fundamentally about modifying the SFT loss. The paper could be tighter here and be more direct. Also, the Off-Policy term here is a bit odd. In RL, off-policy has a specific meaning about learning from data generated by a different policy. Here, off-policy tokens just means low-probability tokens under the current model, which is a different and potentially confusing usage of the term. The experimental section is dense with tables but the key takeaways could be highlighted more effectively since it's pretty straightforward. A summary figure showing aggregated gains would be nice. Would give a 2 here.

Significance: The problem being addressed is genuinely important and is quite critical for current post-training. The SFT-then-RL pipeline is now standard practice, and the tension between adapting to new data and retaining pre-trained (or maybe earlier SFT) knowledge is a real bottleneck. Any method that reliably improves this tradeoff has practical value and the results seem quite meaningful here. The RL improvement numbers are meaningful. A meaningful gain in final RL performance from just changing the SFT initialization is notable and supports the broader point that cold-start quality matters. The downside is the choice of tasks like ALF-World, rather than more critical agentic tasks like SWEBench or BrowseComp, etc. Also, no concrete study of how the initialization change affects RL dynamics -- like why is the large delta appearing? Giving a 3.

Originality: The core idea here is genuinely pretty cool -- the diagnosis that a small subset of tokens is responsible for most of the forgetting during SFT, and that you can selectively constrain just those updates rather than slowing down learning globally. The gradient norm binning experiment is clean and the analysis showing forgetting concentrates early in training adds real value to how SFT goes wrong than what exists currently in literature.  The method itself is PPO clipping with the advantage set to 1, and concurrent work arrives at basically the same thing, so the algorithmic novelty is thin. But the diagnostic framing -- understanding which tokens cause damage and why -- is a legitimate contribution that could inform other approaches beyond just clipping and also the fact we apply it to SFT.  What holds it back is that the whole thing feels a bit contrived in its packaging. The off-policy terminology is doing a lot of heavy lifting and it's not really the right word for what's happening. In RL, off-policy means you're learning from data generated by a different policy. Here we just mean tokens the model assigns low probability to, which is a different thing. A token can be low-probability for lots of reasons -- it could be genuinely novel, or it could be one of several reasonable completions where probability mass is spread thin. Lumping all low-probability tokens together as off-policy and treating them uniformly through a single clipping threshold feels like it's handwaving over meaningful distinctions. The paper would be stronger if it engaged more carefully with what low probability actually signals in different contexts rather than adopting RL terminology that doesn't quite fit. But otherwise, it's very solid. Giving a 3.

---

> ### Author Rebuttal · Authors · 2026-03-31
>
> Dear Reviewer ppip, thank you for your insightful comments. We hope our rebuttal addresses your concerns.
>
> 1. Presentation.
>
> We agree that the RL framing, including state and action space, is unnecessary for understanding our core contribution. In the revised manuscript, we will significantly condense this section and remove the heavy RL notation.
> We also fully understand your concern regarding the non-standard use of the term 'off-policy'. Our original intention was to borrow the RL intuition to describe the divergence between the tokens in the dataset and the model's current behavior. However, we agree that this usage can be confusing given its strict definition in RL. Since 'low-probability' accurately describes the mathematical property of these tokens and perfectly aligns with our observations, we will replace 'off-policy tokens' with 'low-probability tokens' to avoid any misleading implications. Following your valuable suggestion, we will include a summary figure in the revision to better highlight the significant and aggregated performance gains of our method.
>
> 2. Low Probability Tokens Analysis.
>
> We agree that low-probability tokens are not all the same. They may be novel, ambiguous, or task-specific. We further analyze the semantic meaning of the clipped tokens. As shown by the word cloud in [link](https://anonymous.4open.science/r/figure-4A1D/realistic_clip_wordcloud_purple.png), many clipped tokens are not noise; some clearly have task-specific meaning. Our claim is not that they should all be treated as noise or permanently suppressed. Rather, our point is that such tokens can trigger overly aggressive updates at the beginning of training. The model should first learn relatively familiar tokens, and then gradually adapt to unfamiliar but task-critical tokens.
>
> 3. KL Penalty.
>
> We have added a KL-penalty baseline to our experiments. Since the exact KL is not tractable in this setting, we approximate it using the empirical expectation over the dataset during training. Although this regularizes policy updates, our results show that it does not effectively improve out-of-distribution generalization and also hurts learning efficiency. The result is shown in [link](https://anonymous.4open.science/r/figure-4A1D/KL_penalty.png).
>
> 4. More Critical Agentic Benchmark.
>
> We totally agree that stronger agentic benchmarks are important. We choose WebArena as it is already a challenging agentic task for research-scale models in our setting. We show the result below.
>
> Table: Performance on WebArena Benchmark (Before & After RL)
>
> | Backbone | Method | Before RL (Cold-start) | After RL (GRPO) |
> | :--- | :--- | :---: | :---: |
> | **Llama3.2-3B-Instruct** | SFT |  5.91 | 9.24 |
> | | NEFT |  6.16  | 8.62 |
> | | DFT | 5.54 | 10.10 |
> | | OPC-SFT | **7.64** | **12.93** |
> | **Qwen2.5-1.5B-Instruct**| SFT | 5.91 | 8.87 |
> | | NEFT | 5.54 | 9.61 |
> | | DFT |   5.17 | 9.85 |
> | | OPC-SFT | **6.77** | **11.33** |
> | **Qwen2.5-7B-Instruct** | SFT | 17.49 | 24.14 |
> | | NEFT | **20.69**  |  25.86  |
> | | DFT | 18.23 | 24.88 |
> | | OPC-SFT | 18.60 | **28.57** |
>
> Table: OOD Generalization Performance (Before RL)
>
> | Backbone | Method | MBPP | MMLU | HumanEval | GPQA | LiveCodeBench | MATH500 |
> | :--- | :--- | :---: | :---: | :---: | :---: | :---: | :---: |
> | **Llama3.2-3B-Instruct** | SFT | 50.26 | 52.30 | 31.10 | 12.12 | 20.30 | 18.20 |
> | | NEFT | 48.15 | 55.40 | 32.32 | 11.11  | 24.15 | 16.40 |
> | | DFT | 52.38 | 54.10 | **36.59** | 10.10 | 22.10 | 20.80 |
> | | OPC-SFT | **56.08** | **58.80** | 33.54 | **16.16** | **30.12** | **32.20** |
> | **Qwen2.5-1.5B-Instruct**| SFT | 40.48 | 55.10 | 40.24 | 17.68 | 8.20 | 19.80 |
> | | NEFT | 38.10 | 56.80 | 42.68 | 16.67 | 10.10 | 18.40 |
> | | DFT | 42.06 | **58.85** | 41.46 | 18.69 | 9.40 | 21.20 |
> | | OPC-SFT | **46.56** | 56.10 | **46.95** | **20.20** | **12.50** | **24.20** |
> | **Qwen2.5-7B-Instruct** | SFT | 70.90 | 64.10 | 66.46 | 30.81 | 50.12 | 66.80 |
> | | NEFT | 73.02 | 66.50 | 69.51 | **33.33** | 54.20 | 64.20 |
> | | DFT | 69.05 | 65.20 | 67.68 | 28.79 | 52.15 | 68.40 |
> | | OPC-SFT | **78.31** | **70.20** | **71.95** | 29.80 | **59.80** | **74.60** |
>
> 5. RL Exploration Signal.
>
> We have validated that OPC-SFT yields better OOD performance at the starting point, indicating a better initialization. We further investigate the RL training dynamics to examine whether OPC-SFT also affects exploration behavior during RL. We find that the OPC-SFT initialization starts with higher policy entropy, and this higher entropy is maintained throughout RL training. This suggests that OPC-SFT supports more exploratory behavior during RL. The entropy progression is shown in [link](https://anonymous.4open.science/r/figure-4A1D/batch_entropy_comparison.png).

---

> > ### Author Rebuttal · Reviewer_ppip · 2026-04-07
> >
> > Concerns are fully resolved and the new results add more credibility to the results. Thank you! I will raise the score.
> >
> > I am excited to see this applied to more usecases.

---

> > > ### Author Response · Authors · 2026-04-07
> > >
> > > We sincerely thank the reviewer for the encouraging feedback and for raising the score. We are very glad that our rebuttal and the additional results have adequately addressed the concerns. We also appreciate the reviewer's interest in seeing the method applied to more use cases.
> > >
> > > To further support this point, we also test OPC-SFT on base models trained with a 60k subset from the OpenOrca dataset, and evaluate its performance with AlpacaEval 2.0. The results suggest that OPC-SFT remains helpful beyond the agentic setting.
> > > | Backbone | Method | In-Distribution (AlpacaEval 2.0) After RL | | OOD (Code) Before RL | |
> > > | :--- | :--- | :---: | :---: | :---: | :---: |
> > > | | | **LC Win Rate (%)** | **Win Rate (%)** | **HumanEval (%)** | **MBPP (%)** |
> > > | **Qwen2.5-7B** | SFT | 18.4 | 16.2 | 58.0 | 31.7 |
> > > | | NEFT | 19.5 | 17.1 | 61.2 | 33.1 |
> > > | | DFT | 20.2 | 18.5 | 59.5 | 32.3 |
> > > | | OPC-SFT | **21.3** | **19.8** | **63.5** | **34.1** |
> > > | **Llama3.1-8B** | SFT | 21.6 | 18.5 | 48.0 | 35.7 |
> > > | | NEFT | 21.0 | 19.8 | 48.8 | 36.2 |
> > > | | DFT | 20.1 | 19.2 | 48.4 | 36.8 |
> > > | | OPC-SFT | **22.8** | **20.3** | **49.1** | **37.3** |

---

### Decision · Program_Chairs · 2026-04-30

**Decision:**

Accept (regular)

**Comment:**

The authors propose OPC-SFT, which applies PPO-style token-level clipping to bound per-token updates during SFT to deal with the forgetting problem in SFT-then-RL pipelines. Reviewers provided useful feedback, and the authors addressed the reviewers' concerns during the rebuttal.